JCB Journal of Cell Biology

# DDX6 modulates P-body and stress granule assembly, composition, and docking

Nina Ripin[1,2] �***, Luisa Macedo de Vasconcelos[1] ***, Daniella A. Ugay[1] ***, and Roy Parker[1,2] ***

**Stress granules and P-bodies are ribonucleoprotein (RNP) granules that accumulate during the stress response due to the condensation of untranslating mRNPs. Stress granules form in part by intermolecular RNA–RNA interactions and can be limited by components of the RNA chaperone network, which inhibits RNA-driven aggregation. Herein, we demonstrate that the DEAD-box helicase DDX6, a P-body component, can also limit the formation of stress granules, independent of the formation of P-bodies. In an ATPase, RNA-binding dependent manner, DDX6 limits the partitioning of itself and other RNPs into stress granules. When P-bodies are limited, proteins that normally partition between stress granules and P-bodies show increased accumulation within stress granules. Moreover, we show that loss of DDX6, 4E-T, and DCP1A increases P-body docking with stress granules, which depends on CNOT1 and PAT1B. Taken together, these observations identify a new role for DDX6 in limiting stress granules and demonstrate that P-body components can influence stress granule composition and docking with P-bodies.**

## Introduction

Ribonucleoprotein (RNP) granules are membraneless assemblies composed of proteins and RNAs. Cells form multiple types of RNP granules including processing bodies (P-bodies, PBs) in the cytosol, and Cajal bodies, nucleoli, nuclear speckles, and paraspeckles in the nucleoplasm. There are also cell type–specific transport or germ granules and neuronal granules (Buchan, 2014). Moreover, various RNP granules such as stress granules (SGs) can form in response to stress due to the release of mRNAs from polysomes (Kedersha et al., 1999). SGs contain stalled preinitiation complexes, small ribosomal subunits, RNAs, and various RBPs such as G3BPs or PABPs (Protter and Parker, 2016). In contrast, P-bodies contain untranslated mRNAs associated with the mRNA decay machinery including the decapping enzyme DCP1/2, EDC3/4, the 5′–3′ exonuclease Xrn1, and the Lsm proteins (1–7) (Sheth and Parker, 2003). SGs and PBs form docking interactions (Kedersha et al., 2005), which are proposed to facilitate the exchange of components (Anderson and Kedersha, 2008). Docking interactions can be understood as differences in interaction strength between homotypic (e.g., interaction between PB RNPs) and heterotypic (interactions between PB and SG RNPs) RNP granule interactions (Ripin and Parker, 2023).

The formation of RNP granules results from the summation of protein–protein, protein–RNA, and promiscuous intermolecular RNA–RNA interactions (Roden and Gladfelter, 2020; Van Treeck and Parker, 2018), leading to a set of multivalent interactions that allows the assembly of multiple RNPs into a larger RNP granule. In many cases, RNP granule assembly is then driven by the equilibrium binding between these different components, and at the mesoscale, it can be considered a form of liquid–liquid phase separation (Banani et al., 2017; Shin and Brangwynne, 2017).

RNP granules can be regulated through various mechanisms, such as post-translational modifications of RNP granule proteins, protein chaperones, and DEAD-box RNA helicases (Protter and Parker, 2016; Hofmann et al., 2021). One key regulator of stress granule formation is a set of proteins that limit intermolecular RNA–RNA interactions (Budkina et al., 2021; Tauber et al., 2020a, 2020b), referred to as an RNA Chaperone Network (Ripin and Parker, 2022, 2023). One example of an RNA chaperone is the eIF4A RNA helicase that functions as an ATP-dependent RNA binding protein that limits intermolecular RNA–RNA interactions both in cells and in vitro (Tauber et al., 2020a). Other DEAD-box helicases can also function to modulate RNP granule formation, and, in some cases, regulate transitions of RNPs between different RNP granules (Weis and Hondele, 2022; Hondele et al., 2019). Whether other DEAD-box RNA helicases function to limit stress granule formation in a similar manner to eIF4A or through additional mechanisms remains to be determined.

Numerous DEAD-box RNA helicases are found in SGs including DDX1, DDX3X, DDX5, DDX17, and DDX6 (Jain et al.,

---

[1]Department of Biochemistry, University of Colorado Boulder, Boulder, CO, USA; [2]Howard Hughes Medical Institute, Chevy Chase, MD, USA.

Correspondence to Roy Parker: roy.parker@colorado.edu.



2016). Interestingly, the mammalian RNA helicase DDX6 (also known as RCK/p54) is a key factor in PB assembly, and its knockdown induces the disassembly of PBs (Ayache et al., 2015). DDX6 can be observed in SGs (Buchan et al., 2008; Mollet et al., 2008) and was recently suggested to regulate SG maturation (Majerciak et al., 2023). Key issues to understand are how DDX6 affects SG formation, distinguish if SGs are regulated directly by DDX6 or through DDX6's role in PBs, and the cellular mechanisms by which DDX6 could regulate multiple RNP granule formation, disassembly, and interactions between RNP granules.

DDX6 could affect both SGs and PBs in multiple ways. First, DDX6 directly interacts with several translational repression complexes (EIF4ENIF1 [4E-T], PAT1B, LSM14A, and LSM14B), as well as multiple RNA decay factors (DCP1A, DCP1B, EDC3, EDC4, LSM proteins, and CNOT1) (Sharif et al., 2013; Ozgur et al., 2015; Brandmann et al., 2018; Ayache et al., 2015; Kamenska et al., 2016). Moreover, immunoprecipitation experiments revealed RNA-dependent and -independent copurification of SG components with DDX6, including the G3BP proteins, eIF4F, and ATXN2/ATXN2L (Bish et al., 2015; Ayache et al., 2015). DDX6 binds PAT1B, LSM14A, EDC3, or 4E-T using the same surface (Sharif et al., 2013; Ozgur et al., 2015; Tritschler et al., 2009), arguing that a single DDX6 molecule can only bind one of the partners at a time. Thus, DDX6 could act as a scaffold by recruiting components one after the other, promoting intermolecular interactions within a PB.

DDX6 is also known to bind throughout the mRNA. This is supported by CLIP experiments of DDX6, and its yeast ortholog DHH1, which show crosslinking to the mRNA coding sequence, 3′ and 5′ untranslated region (Di Stefano et al., 2019; Mitchell et al., 2013). This suggests that DDX6 could bind throughout mRNAs and then act as an RNA helicase to remodel RNA–RNA or RNA–protein interactions in an ATP-dependent manner, similar to the function of eIF4A in regulating SGs (Tauber et al., 2020a). The remodeling process could exclude RNAs from SGs, and through the interaction with PB components recruit them into PBs.

Another possible mode of action for DDX6 is through remodeling RNA–protein interactions by removing SG proteins from RNAs without RNA duplex unwinding. Indeed, as shown for other DExH/D box proteins (Shibuya et al., 2004; Fairman et al., 2004; Bowers et al., 2006), DDX6 could bind stably to the RNA like a clamp or place holder when bound to ATP. This could prevent RNA-binding proteins from binding mRNAs until the interaction of CNOT1 with DDX6, which promotes ATP hydrolysis and subsequent DDX6 release from RNA (Hondele et al., 2019; Mathys et al., 2014).

Herein, we demonstrate that DDX6 functions largely independently of canonical PBs in reducing SG formation. Current observations suggest that DDX6 acts on limiting RNPs in SGs in a manner dependent on RNA binding and ATPase activity. Moreover, we show that the formation and/or components of SGs and PBs can alter the composition of the other granule as well as affect the docking of PBs and SGs. A general principle that emerges is that reducing the interactions that promote PBs leads to smaller PB-like assemblies that cluster around the periphery of SGs, which can be explained by a weakening of interactions between PB mRNPs leading to increased PB–SG interactions. Taken together, these observations identify a new role for DDX6 in regulating SGs by limiting the partitioning of RNPs, particularly PB and SG components into SGs.

## Results

### P-bodies and SGs increase at different times during stress response with overlapping and unique components

To understand the relationship between PBs and SGs, we first examined their formation over time by exposing osteosarcoma U-2 OS cells to oxidative stress using arsenite (NaAsO$_2$). We then monitored SG and PB formation through immunofluorescence (IF) using the PB and SG markers EDC4 and PABPC1, respectively. Without stress, U-2 OS cells have one to two PBs per cell (Fig. 1, A and B). Upon stress, consistent with earlier results (Buchan et al., 2008; Mollet et al., 2008), we observed PBs increased first, followed by an increase in SGs (Fig. 1, A and B).

Although the protein distribution between PBs and SGs has been extensively studied (Kedersha et al., 2008; Riggs et al., 2020), differences between cell lines were reported in the past (Kedersha et al., 2005; Youn et al., 2018; Hubstenberger et al., 2017; Buchan et al., 2008; Mollet et al., 2008; Kedersha and Anderson, 2007). We, therefore, performed an analysis of the distribution of specific proteins between SGs and PBs in U2-OS cells to assess proteins unique to each granule, determine the degree of enrichment of specific proteins in each granule, and compare localization changes upon certain perturbations that are done in this study. This analysis led to several observations.

First, we observed multiple SG components are also found in PBs including eIF4E, eIF4G1 (Fig. 1, C and D), TIA-1, HuR, UBAP2L, and YB1 (Fig. S1, A and B), in agreement with previous results for eIF4E, eIF4G1, and YB1 (Kedersha and Anderson, 2007; Hubstenberger et al., 2017; Kedersha et al., 2005, 2008; Stoecklin and Kedersha, 2013). However, TIA-1, HuR, UBAP2L, and YB1 had lower partition coefficients (as assessed by mean intensity granule/cytoplasm) into PBs over SGs, demonstrating they are preferentially recruited to SGs (Fig. S1 B). In contrast, eIF4E and eIF4G1 (Fig. 1, C and D) had similar partition coefficients for both PBs and SGs. The detection of SG components in PBs is not due to bleed-through since these SG proteins are also seen in PBs even in the absence of PB antibodies (right panel "IF without PB marker"). TIA-1, HuR, YB1, and UBAP2L also localized in PBs in unstressed conditions (Fig. S1 C).

Second, consistent with earlier results (Kedersha et al., 2005), we observed that PABPC1, G3BP1, and polyadenylated RNAs (as assayed by oligo[dT] RNA FISH) are not enriched in PBs and therefore are markers of SGs (Fig. 1 E).

Third, PB components EDC4, DCP1A, DDX6, 4E-T, and EDC3 are predominantly localized in PBs (Fig. 1, C, E, and F), whereas the PB components LSM4, LSM14A, and CNOT1 also accumulate in SGs (Fig. S1 D). As previously reported in U-2 OS cells (Kedersha and Anderson, 2007), DDX6 is primarily in PB and generally not observed in SGs, although it has been shown in HeLa cells to localize to SGs (Buchan et al., 2008; Mollet et al., 2008). To rule out an artifact of the antibody utilized, a second DDX6 antibody shows the same lack of SG localization in U-2 OS

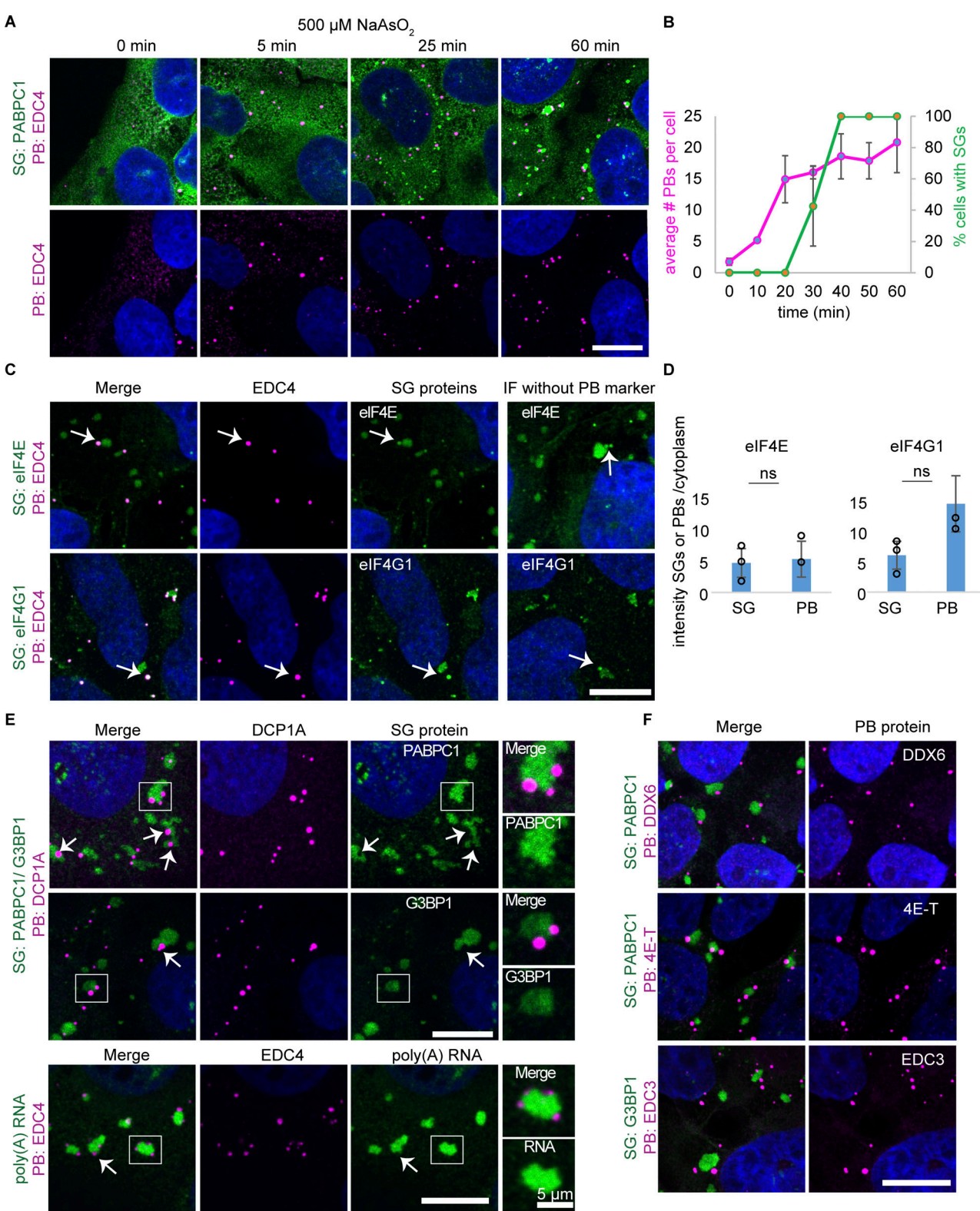

Figure 1. **PBs grow in number before the formation of SGs during stress, contain unique proteins, and share components with SG. (A)** IF images showing the increase in PBs (EDC4 IF, magenta) before the formation of SGs (PABPC1 IF, green) in wild-type U-2 OS cells over a time course after treating cells with 500 µM arsenite. **(B)** Quantification of A displaying the average number of PBs per cell (left axis) and % cells with SGs (right axis). Error bars represent the standard deviation of three independent replicates. **(C)** IF images of eIF4E and eIF4G1 (green) that also colocalize in PBs (EDC4 IF, magenta). Arrows point to examples of PB localization. **(D)** Quantification of mean intensity granule/cytoplasm of proteins in C in SGs and PBs. Each data point represents the mean value of one replicate. Error bars represent the standard deviation of three independent replicates. **(E)** IF images of SG proteins (G3BP1 and PABPC1, green) and poly(A) RNA (oligo[dT] RNA FISH, green) that don't colocalize with PBs (DCP1A or EDC4 IF, magenta). These are neither enriched (examples shown by arrows) nor depleted (examples shown in inset). **(F)** IF images of the PB proteins DDX6, 4E-T, and EDC3 (magenta). IF images are representative images of three

independent biological replicates with more than three images analyzed per replicate. Scale bar, 10 µm if not indicated. ns = not significant (unpaired, two-tailed *t* test).

cells (Fig. S1 E), suggesting cell type–specific differences in DDX6 localization.

Overall, our data is consistent with earlier results that many RBPs can localize to both granules and therefore could be involved in the formation or regulation of both PBs and SGs. In our subsequent experiments, we utilized PABPC1 as our primary SG marker and defined SGs as puncta or larger structures containing PABPC1 and not EDC4, which we used as our primary PB marker. This allows us to define PBs as EDC4 puncta or larger structures lacking PABPC1. These proteins are unique markers for their corresponding granule and are not affected by the protein knockouts and knockdowns performed below.

## DDX6 knockout increases stress granule formation

In addition to DDX6's known role in promoting PB formation (Ayache et al., 2015), we hypothesized that DDX6 might function as an RNA chaperone (Ripin and Parker, 2022, 2023) and limit SG formation for three reasons. First, DDX6 was shown to be much more abundant than other PB components in U-2 OS cells (Fig. S2 A) (Beck et al., 2011) or Hela cells (Ayache et al., 2015; Itzhak et al., 2016). This higher abundance of DDX6 relative to other PB proteins suggested it might have an additional function besides PB assembly. Second, as a DEAD-box helicase, DDX6 is similar to eIF4A, and might therefore limit SG formation in a similar manner (Tauber et al., 2020a). Moreover, DDX6 can localize to SGs based on mass spectroscopy of purified SGs (Jain et al., 2016), IF in HeLa cells (Buchan et al., 2008; Mollet et al., 2008), or specific perturbations in U-2 OS cells (see below).

To test the role of DDX6 in SG formation, we created cell lines lacking DDX6 using CRISPR-Cas9 in U-2 OS wild type and cells lacking G3BP1/2 proteins, which are deficient in SG formation (Kedersha et al., 2016). In both cell lines, the DDX6 knockout cells (DDX6 KO) show a complete loss of DDX6 (Fig. 2 A and Fig. S2 B). We observed a loss in canonical PB formation in unstressed and stressed DDX6 KO and G3BP1/2 + DDX6 triple KO cell lines (Fig. 2 A and Fig. S2, C–E), consistent with the previously described role of DDX6 in PB formation (Ayache et al., 2015). Importantly, all cell lines show no substantial changes in the expression levels of key SG and PB markers used in this study as assessed by Western blot, including G3BP1, PABP DCP1A, and EDC4 (Fig. S2 B). Thus, we have a set of cell lines that, upon stress, make both PBs and SGs (wild-type U-2 OS cells), fail to make SGs but still make PBs (G3BP1/2 KO cells), fail to make PBs (DDX6 KO cells), or fail to make both SG and PB (DDX6 KO+G3BP1/2 KO cells). We used these cell lines to examine how DDX6 affects SG formation, as well as how PBs and SGs affect each other.

To examine how the loss of DDX6 affected SGs, we treated wild-type and DDX6 KO cells with arsenite and examined the formation of SGs. This experiment provided multiple observations suggesting DDX6 limits SG assembly. First, the DDX6 KO cell line formed twice as many SGs per cell as assayed by multiple SG markers (Fig. 2, A–C and Fig. S2 F). As the individual

SGs were on average smaller in DDX6 KO cells (Fig. 2, A, B, and D; and Fig. S2 F), it was possible that loss of DDX6 led to impaired SG growth or fusion. If this is the case, it would cause the same total SG area per cell in both wild-type and DDX6 KO cells. However, the total SG area per cell is almost 2× higher in DDX6 KO cells (Fig. 2 E and Fig. S2 F), consistent with increased SG formation. The differences in SG formation between wild-type and DDX6 KO cells are not due to differences in translation repression since arsenite represses translation similarly in both cell lines (Fig. S2, L and M).

The changes in SG number, individual area, and total SG area per cell can be recapitulated by siRNA knockdowns targeting DDX6 (see below), providing an orthogonal genetic experiment demonstrating a role for DDX6 in limiting SGs. In addition, the increase in SGs in DDX6 KO cell lines is also observed in other stresses such as hippuristanol or sorbitol (Fig. 2 F and Fig. S2 H).

We also observed that in DDX6 KO cells, SGs formed earlier (Fig. 2 G) and persisted longer following recovery after mild stress (Fig. 2 H). This is consistent with DDX6 functioning in some manner to limit SG assembly.

One possibility is that DDX6 promotes the loss of G3BP proteins from mRNAs in SG and thereby limits SG assembly. To test this possibility, we utilized fluorescence recovery after photobleaching (FRAP) to measure the exchange rates of a GFP-G3BP1 in wild-type and DDX6 KO cells. We observed that the DDX6 KO did not alter the dynamics of G3BP1 in SGs (Fig. S2 K), arguing that DDX6 limits SG by an alternative mechanism.

Finally, we examined if the loss of DDX6 could restore SGs in the G3BP1/2 KO cell line. This experiment is based on the logic that SG formation is due to the summation of protein–protein, protein–RNA, and RNA–RNA interactions, and defects in one set of interactions (such as the loss of G3BP proteins) can be rescued by increasing other interactions. For example, prior work has shown that inhibiting eIF4A helicase activity to limit trans RNA–RNA interactions or limiting ADAR1 can restore SG-like foci in G3BP1/2 KO cells (Tauber et al., 2020a; Corbet et al., 2021). To examine if DDX6 acted similarly, we compared SG formation in the G3BP1/2 + DDX6 triple KO cell line to the G3BP1/2 KO cell line. We observed that upon arsenite treatment, the DDX6 KO restored small SG-like foci in the G3BP1/2 KO cells as detected by PABPC1 and eIF4A IF (Fig. 2 I). These small SG-like foci contain various other RBPs such as YB1 and UBAP2L (Fig. S2 I), suggesting they are smaller SGs. This provides an additional observation that DDX6 functions to limit SG formation.

## Loss of DDX6 increases mRNA partitioning into SGs

We previously described eIF4A as a DEAD box helicase that limits SG assembly by competing for intermolecular RNA–RNA interactions (Tauber et al., 2020a). Inhibition of eIF4A also increases the fraction of some mRNAs in SG, suggesting eIF4A limits at least some specific mRNAs from accumulation in SGs. To examine if the loss of DDX6 increases the partitioning of

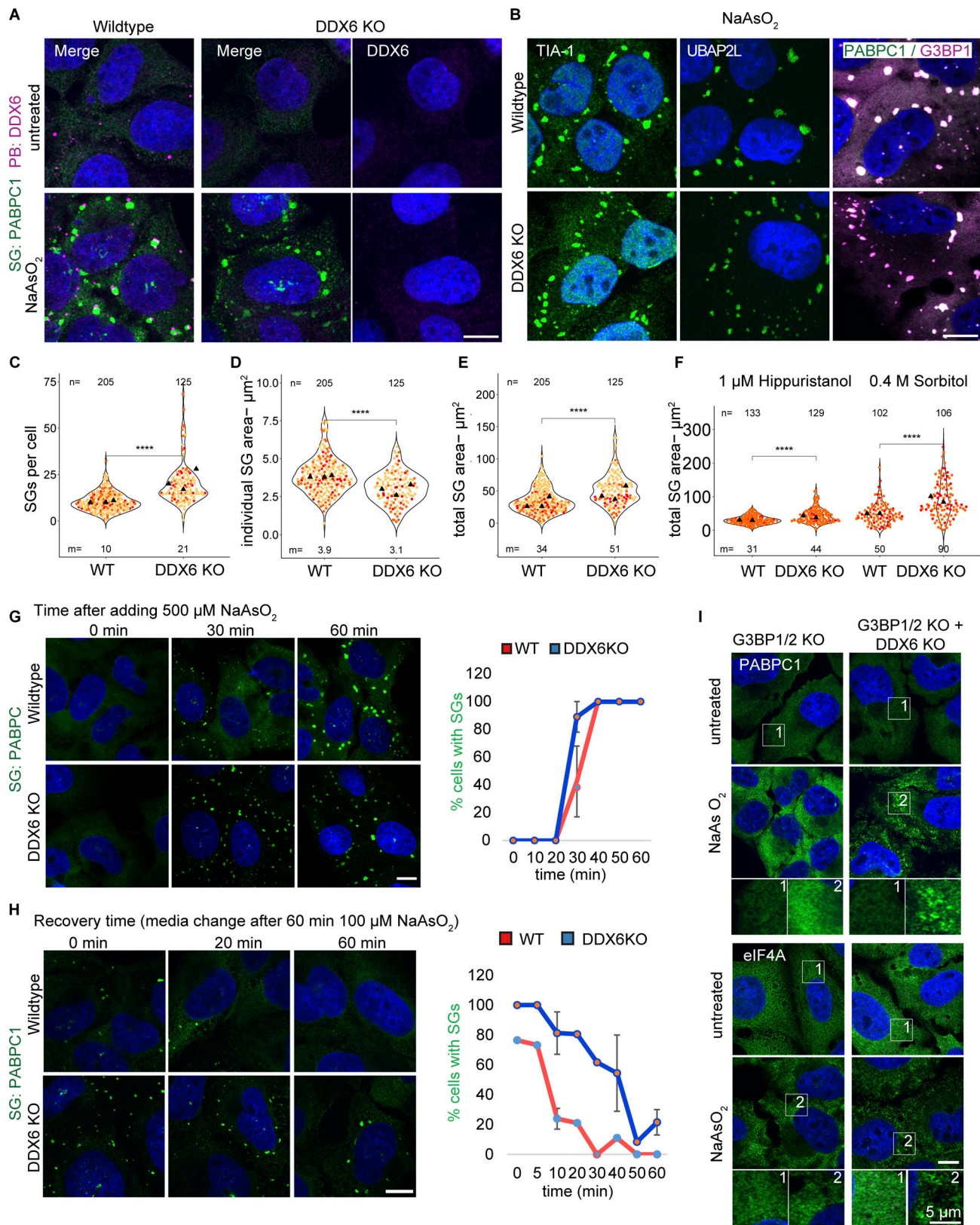

Figure 2. **DDX6 limits SG assembly. (A)** IF images showing the loss of DDX6/PBs (DDX6 IF, magenta) in unstressed (top) and stressed conditions (bottom). **(B)** IF images showing different SG enriched proteins (TIA-1, UBAP2L, and G3BP1). **(C–E)** Using PABPC1 IF images, quantification of SG number, mean SG area, and the total SG area per cell. **(F)** Quantification of the total SG area per cell upon Huppuristanol and Sorbitol stress. **(G)** IF images at 0, 30, and 60 min showed a more rapid increase in SGs (PABPC1 IF, green) in DDX6 KO cells compared with wild-type cells over a time course after treating cells with 500 μM arsenite. On the right is quantification, displaying the % cells with SGs over time. Error bars represent the standard deviation of three independent replicates. **(H)** IF

images showing the faster recovery (PABPC1 IF, green) in wild-type compared with DDX6 KO cells, after treating cells with 100 µM arsenite for 1 h, following media change. On the right is quantification, displaying the % cells with SGs over time. Error bars represent the standard deviation of two independent replicates. **(I)** IF images of PABPC1 (green) in unstressed (top) and stressed (bottom) G3BP1/2 KO and G3BP1/2+DDX6 KO cells (the corresponding wild-type PABPC1 IF is shown in A). Same below but for eIF4A (green). White boxes denote the insets at the bottom. All IF images are representative images of three independent biological replicates with more than three images analyzed per replicate. For quantification in C–E, data points from three and in F from two biological replicates are shown in red-orange-pink, the mean replicate values are indicated as triangles, and mean values are shown on the bottom of the image (m). The number (n) of analyzed cells is shown at the top of the image. ****P ≤ 0.0001, ***P ≤ 0.001, **P ≤ 0.01, *P ≤ 0.05, n.s. P > 0.05 (unpaired, two-tailed t test on individual data points). Scale bar, 10 µm if not indicated.

mRNAs into SGs, we performed single-molecule fluorescence in situ hybridization (smFISH) for two RNAs enriched in SGs (NORAD and DYNCH1), and GAPDH mRNAs, which are depleted from, but still detectable in SGs (Khong et al., 2017). We then quantified the number of mRNAs in both wild-type and DDX6 KO cells and what fraction of those mRNAs were partitioned into SGs.

We observed that a higher percentage of the GAPDH, DYNCH1, and NORAD RNAs were recruited into SGs in the DDX6 KO cells compared with the wild type (Fig. 3, A and B; and Fig. S3 A). We also observed that upon DDX6 KO, the total number of GAPDH mRNAs decreases, whereas DYNCH1 and NORAD RNAs were elevated (Fig. S3 B), presumably because of DDX6 effects on mRNA turnover (Cruchez et al., 2019; Rouya et al., 2014). The increased accumulation of mRNAs in SGs in the DDX6 KO cell line is consistent with DDX6 playing a role in limiting RNA recruitment and SG formation.

### P-body-like assemblies cluster in and around SGs in DDX6 KO cells

In DDX6 KO cells under stress conditions, PBs are largely abolished (Fig. 2 A; and Fig. S2, C and D), as expected from the role of DDX6 in PB assembly (Ayache et al., 2015). However, we observed that some PB components such as EDC4, DCP1A, and 4E-T form smaller puncta that colocalize in, and mostly line the periphery of SGs (Fig. 4 A), which resembles the docking of PBs to SGs (Kedersha et al., 2005). Similarly, G3BP1/2 + DDX6 triple KO cells show increased interaction or colocalization of the smaller SGs with EDC4 puncta (Fig. S2 J). This suggests that even in the absence of DDX6, smaller PB-like assemblies can form under stress conditions, and those assemblies preferentially dock to the

surface of SG. This is similar to results in yeast where PBs are assembled through a diversity of different protein–protein interactions, and removing one component leads to the formation of smaller PB assemblies, sometimes below the detection of light microscopy (Rao and Parker, 2017). We now refer to the large, mature PBs that form in wild-type cells as canonical PBs to distinguish them from smaller PBs or PB-like assemblies that are caused by perturbations.

The formation of smaller distinct PB-like assemblies seemed to first contradict previous observations, where, upon loss of DDX6, some PB components show colocalization with SGs (Majerciak et al., 2023; Serman et al., 2007; Mollet et al., 2008). However, we noticed that some of those previous images also showed punctate signals of PB components of stronger intensity at the SG periphery. One plausible explanation for these different observations could be differences in image resolution and reduction of out-of-focus light allowing for a better resolution of the punctate signal.

To better resolve the interaction of smaller PBs with SGs, we investigated the localization patterns using the Nikon Spinning Disk Super Resolution by Optical Pixel Reassignment (SoRa) microscope, which improves resolution and reduces out-of-focus light. 3D images are visualized, and surface objects were added using the Imaris Cell Imaging Software (Fig. S3 C). This 3D visualization reveals that the majority of EDC4 and DCP1A puncta are indeed docking to the SG surface, with only a few puncta that can be seen inside (Fig. 4 B, left and center). In contrast, puncta of the 4E-T protein are more heterogeneous and located at the SG center and inside (Fig. 4 B, right, E). Moreover, imaging at higher magnification reveals that EDC4 and 4E-T are predominantly forming distinct subassemblies, with only

**Figure 3.** **Upon loss of DDX6, RNA enrichment is increased in SGs. (A)** IF images comparing NORAD RNA localization in SGs in wild-type and DDX6 KO cells. Quantification of % NORAD RNA in SGs is shown on the right. **(B)** Quantification of % DYNCH1 and GAPDH RNA in SGs. All IF images are representative images of three independent biological replicates with more than three images analyzed per replicate. For quantification, data points from three biological replicates are shown in red-orange-pink, the mean replicate values are indicated as triangles, and mean values are shown at the bottom of the image (m). The number (n) of analyzed cells is shown at the top of the image. ****P ≤ 0.0001, ***P ≤ 0.001, **P ≤ 0.01, *P ≤ 0.05, n.s. P > 0.05 (unpaired, two-tailed t test on individual data points). Scale bar, 10 µm if not indicated.

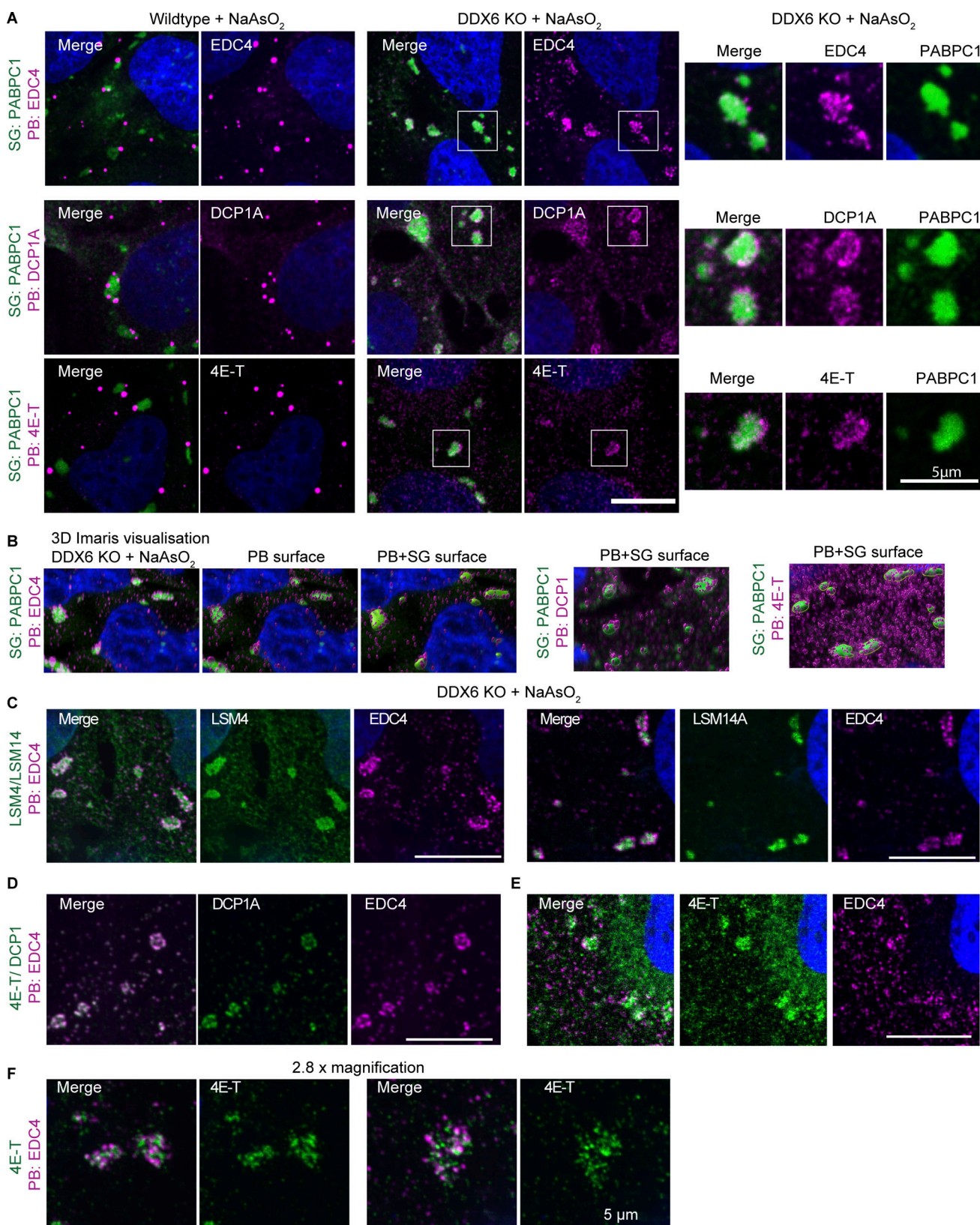

Figure 4. **In the absence of DDX6, smaller PB-like assemblies colocalize or dock to SG. (A)** IF images comparing EDC4, DCP1A, and 4E-T (magenta) PB localization in wild type and redistribution or docking to SGs (green) in DDX6 KO cells. Corresponding zoom images of selected SGs (indicated by white box) are shown on the right. **(B)** SoRa images of stressed DDX6 KO cells visualized in 3D using Imaris. Left: PABPC1 (green) and EDC4 (magenta) z-stacks (left), with added PB surfaces (center) and PB–SG surfaces (right). PABPC1-DCP1A or PABPC1-4E-T PB–SG surface 3D images are shown in middle and right. **(C)** SoRa IF images highlighting SG localization of PB proteins LSM4 and LSM14 in DDX6 KO cells. **(D)** SoRa IF images depicting EDC4 and DCP1A colocalization. **(E)** SoRa IF

images depicting different localization patterns between 4E-T and EDC4. **(F)** SoRa IF images but with 2.8× magnification of 4E-T and EDC4 assemblies. All IF images are representative images of three independent biological replicates with more than three images analyzed per replicate. Scale bar, 10 μm if not indicated.

limited colocalization. We hypothesize that 4E-T preferentially forms different subassemblies from DCP1A and EDC4 in DDX6 KO cells due to the loss of DDX6-4E-T interactions that stabilize 4E-T in PBs.

We examined the localization patterns of additional PB components in DDX6 KO cells using SoRa. LSM4 and LSM14 show a rather uniform signal within SGs, surrounded by EDC4 puncta (Fig. 4 C). This is not surprising as both proteins are already enriched or localized in SGs in wild-type cells demonstrating they have an affinity for SG components (Fig. S1 D). Similarly, eIF4E, eIF4G, and CNOT1 show enrichment in both SGs and PBs in wild-type cells (Fig. 1 C and Fig. S1 D) and also show a relocation from PBs into SGs in a diffuse manner upon loss of DDX6 (Fig. S3 D). A similar localization pattern was recently observed for GW182 in HeLa cells, where, upon DDX6 loss, GW182 largely relocalizes into SGs as well as forming punctate signal at the SG periphery (Majerciak et al., 2023).

Our observations reveal two distinct PB component localization patterns upon loss of DDX6. PB components that are found within SGs in wild-type cells, relocalize, or "mix" better into SGs. We interpret this to occur because these proteins have interactions with SG components that lead to their accumulation in SG in wild-type cells. In DDX6 KO cells, the reduction in canonical PBs reduces competition with PB assembly, leading to their accumulation in SG. In contrast, PB components that do not accumulate in SG in wild-type cells form distinct puncta at the SG periphery or subassemblies within SGs. We interpret this pattern to occur because those proteins can still form small PBs, which then have increased interactions with SGs and hence form puncta associated with SG. Some PB components show intermediate phenotypes, as observed for 4E-T (Fig. 4, E and F) and GW182 (Majerciak et al., 2023).

Overall, we suggest that EDC4, DCP1A, and some 4E-T puncta represent subassemblies of PBs, hereafter referred to PB-like assemblies, since EDC4 and DCP1A fully colocalize in these puncta (Fig. 4 D), while only a few 4E-T puncta can colocalize with EDC4 (Fig. 4, E and F). We interpret these patterns of PB protein localization (EDC4/DCP1A versus 4E-T) to represent two protein–RNA subassemblies that might be coassembled through DDX6 into a wild-type PB, but in the absence of DDX6 partition differentially into SGs. This is consistent with a model whereby the partitioning of complexes into SG or PB is dependent on the strength of competing homotypic versus heterotypic RNP granule interactions (Sanders et al., 2020; Ripin and Parker, 2023). Loss of DDX6 leads to the formation of smaller PB assemblies that now have an increased propensity to dock with SG.

### DDX6 limits SGs independent of P-bodies or P-body-like assemblies

Multiple models could explain the increase in SGs in the DDX6 KO cell lines. First, it is conceivable that apparent SG formation is increased solely due to the docking of PB-like assemblies with

SG in the DDX6 KO cells (Fig. 4) and is independent of a direct DDX6 function in limiting SGs. This model predicts the same total collective PB and SG area in wild-type and DDX6 KO cells. To address this possibility, both PB and SG channels were merged (Fig. S2 G, left), and the overall area was determined on the newly generated merged SG+PB surfaces (Fig. S2 G, right). As for the original SG-only measurements, DDX6 KO cells still form more assemblies (Fig. S2 G right versus Fig. S2 F). This analysis rules out that the change in SGs upon DDX6 KO is only due to the addition of smaller PB-like assemblies into SGs.

In response to stress, PBs form earlier and disassemble later (Fig. 1 A) (Buchan et al., 2008; Mollet et al., 2008). Therefore, since SGs in the DDX6 KO cells contain PB proteins or interact with PB-like assemblies, the difference in the SG assembly and disassembly rates in DDX6 KO cell lines (Fig. 2, G and H) could be due to the smaller PB-like assemblies forming first and then seeding more SGs. However, compared with wild-type cells, where PBs form earlier, no PB-like assemblies can be observed at earlier points in DDX6 KO cells (Fig. S3 E, 10 min). In DDX6 KO cells, small SGs first appear after 20 min with PB-like assemblies docking with these SGs (Fig. S3 E, 20 min). Since some SGs at 20 min in DDX6 KO cells show less interaction with EDC4 puncta (indicated by arrow), this suggests that SG formation is followed by the recruitment of smaller PB-like assemblies. These observations exclude a possible function of the smaller PB-like assemblies in DDX6 KO cells promoting increased SG formation. This time course data also indicates that while DDX6 is needed for canonical PBs, it is not needed for the assembly of SGs or promoting interactions between SGs and EDC4 puncta.

### Other P-body components can affect stress granule formation in DDX6-independent and -dependent manners

In principle, the increased SGs seen in DDX6 KO cells could be unique to DDX6 or could be influenced by other PB components. To address this possibility, we examined how siRNA knockdowns of other PB components affected SG formation as assessed by changes in SG total area (which integrates SG number and size). Knockdowns of PB components were validated by Western blotting with knockdown efficiencies of around 66% (CNOT1), 80% (DCP1A), 83% (EDC3), 100% (4E-T), 80% (LSM14A), 78% (PAT1B), and 83% for DDX6 itself compared with non-targeting control siRNA (Fig. S4 A).

We observed that the siRNA knockdown of PB components leads to three main phenotypes, largely in agreement with a prior study in HeLa cells (Ayache et al., 2015). First, the knockdown of DDX6 recapitulates the phenotypes of DDX6 KO cells, showing a loss of canonical PBs and an increased number of smaller PBs during arsenite treatment (Fig. 5, A and B; and Fig. S4 B, highlighted by blue box). A similar phenotype is observed for 4E-T knockdown, due to the importance of the 4E-T-DDX6 interaction for PB assembly (Kamenska et al., 2016). DCP1A knockdown showed a minor increase in PB numbers

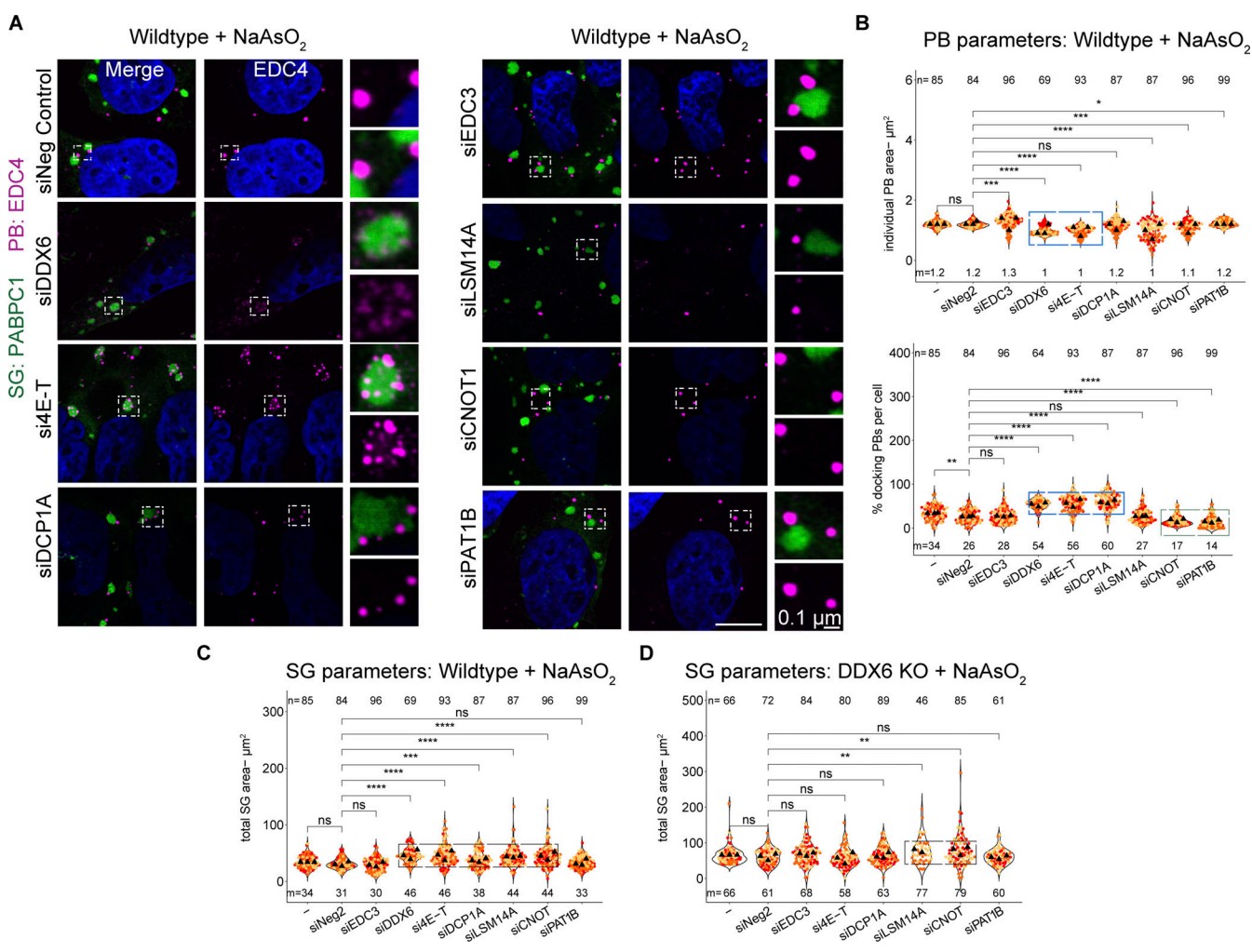

Figure 5. **siRNA knockdowns of various PB components lead to an increase in SGs, docking of smaller PBs, or loss of PB–SG docking. (A)** IF images of PBs (EDC4 IF, magenta) and SGs (PABPC1, green) in wild-type cells upon siRNA knockdown of various PB proteins. **(B)** Quantification of total PB area per cell and % docking interactions with SGs in wild-type cells. Blue boxes highlight conditions that affect the size of PBs (top) that increasingly dock to SGs (bottom). The green box highlights conditions that lead to reduced PB–SG docking. **(C)** Quantification of total SG area per cell in wild-type cells. Black boxes highlight conditions that lead to an increase in SGs. **(D)** Quantification of total SG area per cell in DDX6 KO cells. Black boxes highlight conditions that lead to an increase in SGs. All IF images are representative images of three independent biological replicates with more than three images analyzed per replicate. For quantification in B–E, data points from three biological replicates are shown in red-orange-pink, the mean replicate values are indicated as triangles, and mean values are shown at the bottom of the image (m). The number (n) of analyzed cells is shown at the top of the image. ****P ≤ 0.0001, ***P ≤ 0.001, **P ≤ 0.01, *P ≤ 0.05, n.s. P > 0.05 (unpaired, two-tailed t test on individual data points). Scale bar, 10 μm if not indicated. The non-targeting control siRNA Pool #2 was abbreviated with siNeg Control or siNeg2. Columns marked as "–" are untransfected cells.

(Fig. 5, A and B; and Fig. S4 B, highlighted by blue box), although it was shown not to affect PBs in HeLa cells (Ayache et al., 2015), highlighting cell type–specific differences. Finally, knockdown of CNOT1, PAT1B, LSM14A, and EDC3 showed no or only minor changes in PB number and individual PB area (Fig. 5, A and B; and Fig. S4 B) (LSM14A knockdown shows only a reduction in size, although it was shown to be essential for PBs in other cell types [Ayache et al., 2015; Di Stefano et al., 2019], while EDC3 and PAT1B knockdown shows a slight reduction in numbers, although it was shown not to affect PB numbers [Ayache et al., 2015], again highlighting cell type–specific differences). This diversity of PB phenotypes allowed us to examine if there were consistent correlative changes in SGs under these conditions.

An interesting result was that SG formation in the knockdowns of PB components failed to show a correlation between the effects on SGs and PBs. Specifically, we observed that knockdowns of DDX6, 4E-T, and DCP1A show different effects on PB formation, but all showed an increase in total SG area per cell (Fig. 5 C). Moreover, knockdown of LSM14A and CNOT1 also showed a similar increase in SG formation, even though PBs were similar to wild type. This argues that the increase in SG with DDX6 knockdowns is not simply due to the reduced levels of PBs and identifies other components of PBs that can affect SG formation.

In principle, the increase in SGs in the 4E-T, DCP1A, CNOT1, and LSM14A knockdowns could be independent of DDX6 function or due to their direct protein interactions with DDX6 that

could alter DDX6 function in a manner leading to an increase in SG. To determine if any of the 4E-T, DCP1A, CNOT1, or LSM14A proteins could affect SG independent of DDX6, we examined the effect of their knockdowns in DDX6 KO cells. We observed that 4E-T and DCP1A knockdowns show no additional change in SG formation compared with the level seen in the DDX6 KO cells (Fig. 5 D). This argues that the effect of 4E-T and DCP1A knockdown in increasing SG is dependent on DDX6 protein. In contrast, the CNOT1 and LSM14A knockdowns increased SGs even in the DDX6 KO cell line to a small, but statistically significant, effect (Fig. 5 D, highlighted by black box). This suggests that CNOT1 and LSM14A influence SG formation independent of DDX6 protein.

Taken together, these results show that alterations in multiple PB components can affect the formation of SGs in both DDX6-dependent and -independent manners, but not in a manner directly related to canonical PBs.

### P-body components can promote or limit P-body-stress granule docking

The docking of PBs and SGs is modulated by interactions between molecules on the surface of PBs and SGs (Sanders et al., 2020). Given this, knockdowns of proteins within PBs might be expected to decrease PB–SG docking when that protein interacts with components of SGs or potentially could increase PB–SG docking when the PB component limits other PB–SG interactions. To identify the impact of PB proteins in PB–SG docking, we examined how the knockdowns on PB proteins affected PB–SG docking.

The DDX6 knockdown recapitulated the accumulation of small PB-like assemblies around SGs seen in the DDX6 KO cells (Fig. 5, A and B). Similarly, the knockdown of 4E-T led to increased docking of small PBs with SG (Fig. 5, A and B). Interestingly, although PBs were not reduced in the DCP1A knockdown, there was an increase in PB–SG docking (Fig. 5, A and B), which argues that increased docking is not solely a consequence of smaller PBs.

We also performed the same siRNA knockdowns in DDX6 KO cells (Fig. S4 D) to test for DDX6-specific effects. Here, we counted the number of EDC4 spots in the cytoplasm and in SGs and determined the percentage of spots within or docked to SGs (Fig. S4 E). Indeed, 4E-T and DCP1A knockdown show no additional effects on docking EDC4 spots in the DDX6 KO cells (Fig. S4 E, highlighted by blue box). This argues that DDX6, DCP1A, and 4E-T all act in a concerted mechanism to limit PB–SG docking.

Additional knockdowns argued that CNOT1 and PAT1B can promote PB–SG docking. Specifically, we observed that knockdown of CNOT1 or PAT1B reduced the docking of PB and SG both in U-2 OS wild-type cells (Fig. 5, A and B [bottom, highlighted by green box]) and reduced the docking of EDC4 spots to SG in DDX6 KO cells (Fig. S4, E and F, highlighted by green box). These observations suggest that CNOT1 and PAT1B promote SG–PB docking, possibly through specific protein–RNA or protein–protein interactions. Moreover, although the docking of PBs or EDC4 spots to SGs in wild-type or DDX6 KO cells, respectively, is reduced with the knockdown of CNOT1 or PAT1B, only the

CNOT1 knockdown shows increased SGs in both cells (Fig. 5, C and D). This highlights that the increase in SGs upon CNOT1 knockdown is independent of the decrease in PB–SG docking.

### DDX6 mutants that impair binding to key P-body components can rescue P-bodies that still increasingly dock to SGs

To gain insight into how DDX6 limits SGs, we constructed a series of point mutations in DDX6, based on prior work on DDX6, or the highly conserved yeast ortholog DHH1. All generated mutations are conserved between both proteins (Fig. S5 A) and were shown to alter specific protein or RNA interactions or reduce the ATPase activity (Minshall et al., 2009; Dutta et al., 2011; Sharif et al., 2013; Mathys et al., 2014; Ozgur et al., 2015; Kamenska et al., 2016; Brandmann et al., 2018; Balak et al., 2019) (Table 1; and Fig. 6, A and B). We stably expressed these mutants in the DDX6 KO cell line using lentivirus transduction of the DDX6 gene and validated the expression of the DDX6 protein in all constructs via Western blotting or IF (Fig. 6, C and D; and Fig. S5 C). We then examined how these DDX6 mutations affected PB (using two different markers) and SG formation, as well as PB–SG docking.

These experiments led to the following results. First, we observed that several mutants failed to restore DDX6's ability to generate canonical PBs. Specifically, DDX6 Mut3 (S343D, Q345D, R346D, altered protein and RNA binding), Mut 4 (R373A, T391A, R421A, reduced RNA binding), E247A (ATPase dead mutant), and R386E (inhibits interaction with CNOT1) (Table 1) mutants cannot restore PBs (Fig. 6 D and Fig. S5 E). A requirement for DDX6 ATPase activity and RNA binding for PB assembly is consistent with earlier results (Minshall et al., 2009; Dutta et al., 2011; Ozgur and Stoecklin, 2013; Mugler et al., 2016).

Second, we observed that wild-type DDX6, Mut1 (Q320A, H323A, T327A, R331A), and Mut2 (R443A, F444A, K447A, E450A, and E451A) (Table 1) rescued PBs in stressed cells (Fig. 6 C and Fig. S5 E). In unstressed cells, it is difficult to judge the extent of PB rescue when reintroducing DDX6 since only a few wild-type cells have PBs (Fig. 1 and Fig. S5 D). However, as a few PBs can be observed (Fig. S5 D), this is indicative of a PB rescue. Mut1 and Mut2 show an increase in PBs (Fig. S5 D) compared with wild-type cells, potentially due to impaired decapping and accumulation of capped untranslated RNAs, as the PB size and number also increase in yeast or mammals when 5′–3′ mRNA decay is decreased (Cougot et al., 2004; Sheth and Parker, 2003). This contrasts with the partial rescue by Mut1 observed in HeLa cells (Ayache et al., 2015), consistent with differences between cell lines and/or DDX6 expression levels. Third, upon stress, Mut1 and 2 showed more PBs with increasing docking compared with DDX6 wild type (Fig. 6, C and E; and Fig. S5 E). This phenotype is similar to the 4E-T knockdown and could be due to both mutations affecting 4E-T binding. The assembly of PBs in Mut1 and Mut2 argues that DDX6 binding to EDC3, PAT1B, 4E-T, and LSM14A is not required for de-novo PB formation but for increasing multivalency between PB RNPs.

After identifying DDX6 mutations that can rescue PBs (Mut1+2) or prevent PBs (all other mutations), we tested how those mutations affected SG formation when expressed in the DDX6 KO cells. As expected, expression of DDX6 wild type

Table 1.  Summary of generated DDX6 mutants, their published phenotypes and our observations

| Mutation | Published effects on DDX6 interactions | Published impacts on PB formation | Observed impact on PBs or SGs in this study |
|---|---|---|---|
| Mut1 (Q320A, H323A, T327A, R331A) | Impairs binding to EDC3-DHH1 (Sharif et al., 2013), 4E-T-DDX6 (Ozgur et al., 2015), probably LSM14A (Brandmann et al., 2018) and weakens PAT1B-DDX6 (Sharif et al., 2013) binding | Two times less potent than DDX6wt at assembling P-bodies (Ayache et al., 2015) | Rescues PBs (increased number and docking), weak limitation of SGs |
| Mut2 (R443A, F444A, K447A, E450A, E451A) | Impairs EDC3-DHH1 (Sharif et al., 2013), then probably LSM14A (Brandmann et al., 2018) but can still bind PAT1-DHH1 (Sharif et al., 2013) | - | Rescues PBs (increased number and docking), increased SGs |
| Mut3 (S343D, Q345D, R346D) | Impairs DDX6-PAT1B binding (Sharif et al., 2013), 4E-T (Ozgur et al., 2015), and most likely LSM14A (Q345 and R346 form three side specific H bonds with LSM14A) thus mutation is expected to abolish or decrease binding to LSM14A (Brandmann et al., 2018), can still bind DDX6-EDC3 (Sharif et al., 2013). Mutation of the same helix with one overlapping residue (R346A, K352A, K353A) impairs PAT1B, DCP2, AGO2 and TNRC6A binding (Ozgur and Stoecklin, 2013). Potentially reduced/abolished RNA binding[a] | No PBs with R346A, K352A, K353A mutant in HeLa cells (Ozgur and Stoecklin, 2013) | No PBs, increased SGs |
| Mut4 (R373A,T391A R421A) | Impairs RNA binding (Dutta et al., 2011), and ATP hydrolysis for DHH1 (Dutta et al., 2011), reduce binding of DDX6 to 4E-T and PAT1B and abolishes binding for LSM14A (Balak et al., 2019) | R322A, S340A, R370A in DHH1: Reduction of PBs (Mugler et al., 2016), R322A/S340A in DHH1: Smaller PBs (Dutta et al., 2011), R373Q or T391I missense mutations in DDX6 abolish PBs in patient cells (Balak et al., 2019) | No PBs, increased SGs |
| E247A | Impairs ATPase activity but is capable of RNA and ATP binding (Dutta et al., 2011) | D195A/E196A in DHH1: Increase in size/number of PBs (Dutta et al., 2011), E195Q in DHH1: Constitutive PBs (Mugler et al., 2016) | No PBs, increased SGs |
| R386E | Impairs CNOT1 binding (Mathys et al., 2014). As CNOT1 can stimulate the RNA-dependent ATPase activity (Mathys et al., 2014), we expect this mutant to behave similarly to the ATPase inactive E247A mutant | R55E, F62E, Q282E, N284E, R355E in DHH1: Constitutive PBs (Mugler et al., 2016) | No PBs, increased SGs |

[a]Like previous observations in DHH1 (Sharif et al., 2013), we noticed a positively charged patch within this mutation containing helix (Fig. S5 B), suggesting partial RNA binding within this area. Our mutations to negatively charged residues could therefore significantly impair RNA binding. Moreover, all our tested parameters resemble the RNA binding incapable Mut4.

rescues the SGs number, individual SG area, and total SG area per cell to the same levels as in wild-type cells (Fig. 6 F). Strikingly, Mut2 still gives increased SGs in the DDX6 KO cells, while Mut1 partially rescues the SGs phenotypes (Fig. 6 F). Since both Mut1 and Mut2 allow for PB formation, this difference provides additional evidence that DDX6 affects SG formation independently of PB formation. The results for Mut1+2 are in agreement with the siRNA knockdowns, where 4E-T and DCP1A knockdowns lead to more PBs that increasingly dock to SGs. Moreover, none of the other DDX6 mutants can rescue SGs to the DDX6 wild-type phenotype (Fig. 6 F).

These observations have three implications. First, it argues that RNA binding is important for DDX6 to limit SG formation since Mut3 and Mut4, which both reduce RNA binding, do not allow DDX6 to limit SG formation (Fig. 6, D and F). Second, it shows that ATPase activity is important for DDX6 in limiting SG formation, as the ATPase-dead mutant E247A, and the R386E mutation, which prevents CNOT1 binding and ATPase

stimulation (Mugler et al., 2016), both fail to limit SG formation. Lastly, abolished binding of key PB components impairs DDX6 in limiting SGs, highlighting the importance of DDX6 binding partners.

## DDX6 defects lead to the increased accumulation of DDX6, G3BP1, and other proteins in SGs

The requirement for RNA binding and ATPase activity for DDX6 to limit SG formation suggested that DDX6 might limit SG by binding RNAs that accumulate in SG, then utilize ATP hydrolysis to promote protein and/or RNA release from SG. This hypothesis predicts that DDX6's accumulation in SGs should be reduced by Mut3 and Mut4, which limit DDX6-RNA binding, and increased by the E247A and R386E mutations, which limit ATPase hydrolysis. Strikingly, quantification of the DDX6 signal in the cytoplasm versus SGs revealed enrichment of DDX6 itself in SGs upon ATPase mutation (E247A, R386E) and a reduction in DDX6 partitioning into SG with the RNA binding mutants (Mut3 and

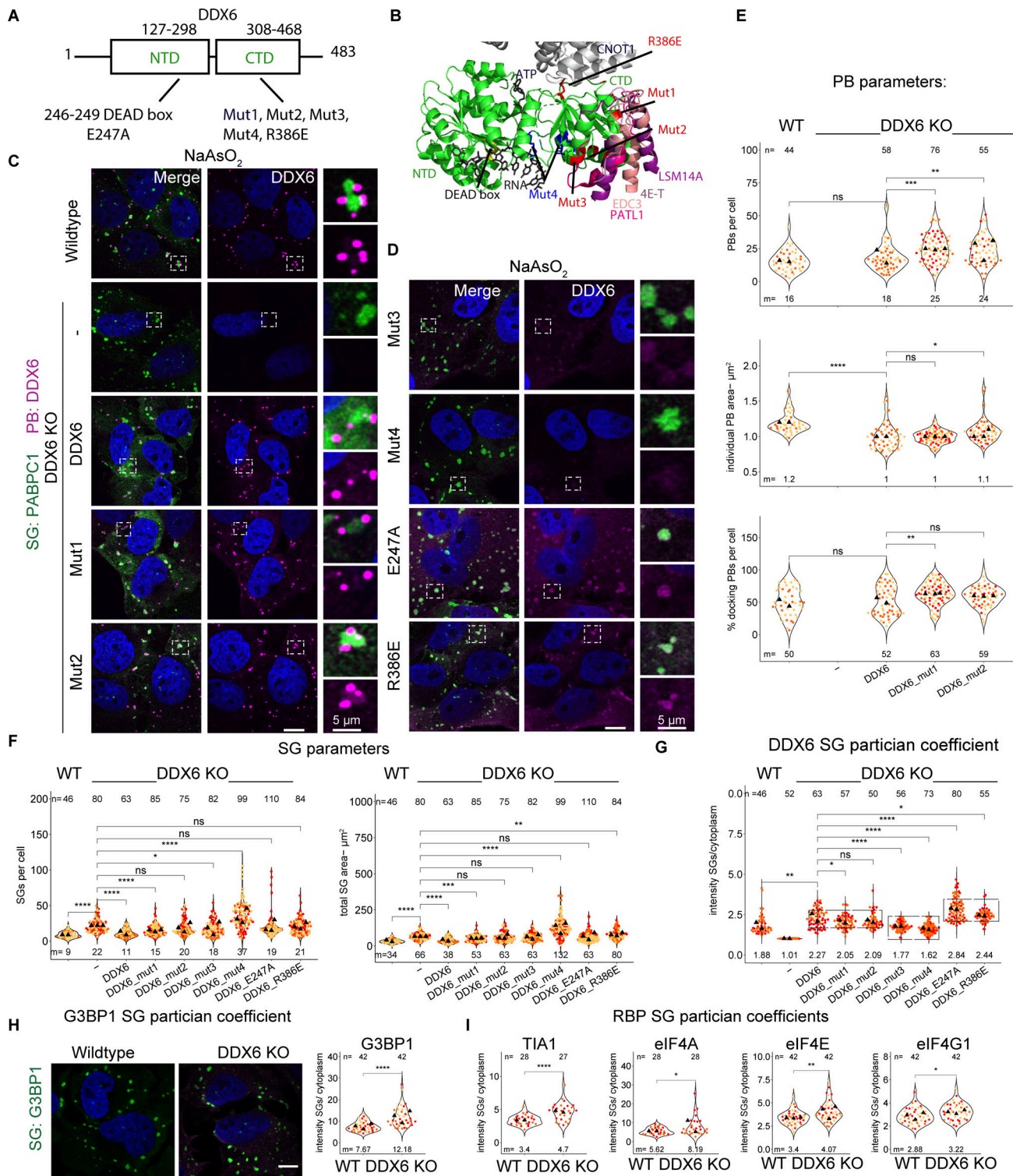

Figure 6. **DDX6 mutants show different PB phenotypes, increased DDX6 partitioning in SGs, and no effect on limiting SGs. (A)** Schematic diagram of DDX6 domains and locations of mutations. **(B)** Structure of a human 4E-T/DDX6/CNOT1 complex (PDB accession no. 5anr) is overlayed with various DDX6 or yeast ortholog DHH1 binding partners indicated with different colors using Pymol (yeast EDC3 peptide from the DHH1-EDC3 structure [PDB accession no. 4bru], RNA/ATP from the DBP5-RNA structure [PDB accession no. 3fht] yeast PAT1 peptide from DHH1-PAT1 structure [PDB accession no. 4brw]). Mutations are highlighted in red (Mut1-3, R386E) and blue (Mut4). **(C)** IF images of PBs (DDX6 IF, magenta) and SGs (PABPC1, green) in DDX6 KO cells stably transduced with DDX6 and various DDX6 mutants that rescue PBs. **(D)** Same as C but with DDX6 mutants that do not rescue PBs. **(E)** Quantification of PB number, total PB area per cell and % docking interactions with SGs. **(F)** Quantification of total SG number and total SG area per cell in wild-type cells. **(G)** Quantification of the mean intensity DDX6 SG/cell. The boxes group three phenotypes (ratio in PB formatting cells, the decreased ratio in RNA binding mutants, and the increased ratio in ATPase inactive mutants. **(H)** IF images of G3BP1 (green) enrichment in wild-type and DDX6 KO cells (left) and quantification of G3BP1 protein mean intensity in SG/cytoplasm (right). **(I)** Quantification of TIA-1, eIF4A, eIF4E, and eIF4G1 intensity in SG/cytoplasm in wild-type and DDX6 KO cells. All IF images

are representative images of three independent biological replicates with more than three images analyzed per replicate. For quantification in B–I, data points from two to three biological replicates are shown in red-orange-pink, the mean replicate values are indicated as triangles, and the mean values are shown on the bottom of the image m). The number ($n$) of analyzed cells is shown at the top of the image. ****$P \leq 0.0001$, ***$P \leq 0.001$, **$P \leq 0.01$, *$P \leq 0.05$, n.s. $P > 0.05$ (unpaired, two-tailed $t$ test on individual data points). Scale bar, 10 μm if not indicated. Columns marked as "–" are untransduced cells.

Mut4) (Fig. 6, D and G). Consistent with this hypothesis, we observed that various other SG and PB components showed increased localization in SGs upon loss of DDX6, such as G3BP1 (Fig. 6 H), TIA-1, eIF4A, eIF4E, eIF4G1, (Fig. 6 I) and LSM14A, LSM4 (Fig. S5 E and Fig. 4 B) while HuR, YB1, and UBAP2L were not affected (Fig. S5 F).

Overall, we interpreted that these results suggest that DDX6 is recruited to SGs in a RNA-dependent manner and is released by ATP hydrolysis, which remodels the SGs in a manner that reduces SG formation (Fig. 7).

## Discussion

We provided four observations that DDX6 limits the formation of SG. First, DDX6 KO cell lines form SGs faster and assemble more SGs than wild-type cells (Fig. 2, A–G; and Fig. S2, F and G). Second, and consistent with DDX6 limiting SG formation, we observed that DDX6 KO cells show slower SG disassembly than wild-type cells (Fig. 2 H). Third, we observed that in DDX6 KO cells, a higher fraction of specific mRNAs and proteins partition into SG (Figs. 3, 4, and 6). Finally, we observed that SG formation can be partially restored by the DDX6 KO in G3BP1/2 double-knockout cell lines (Fig. 2 I and Fig. S2 I). This suggests that DDX6 is another RNA chaperone (Tauber et al., 2020b; Ripin and Parker, 2022, 2023) that regulates the propensity of RNP assembly into SGs.

Several observations argue that DDX6 does not limit SG formation simply by promoting the formation of PBs. First, the effects of specific mutations in DDX6 on PB formation and limiting SG do not correlate (Fig. 6). For example, the expression of Mut1+2 in DDX6 KO cells rescues PBs but does not fully reverse the SG phenotypes of the DDX6 KO cells (Fig. 6, C, E, and F). Second, knockdowns of other PB components such as CNOT1 and LSM14A can increase SG, but without altering PB formation (Fig. 5). Finally, while 4E-T and DCP1A knockdowns resemble DDX6 KO cells the most in terms of numbers and docking, PBs are still present, although altered (Fig. 5, A–C).

Based on the requirement for RNA binding and ATP hydrolysis to limit SG formation, we suggested that DDX6 limits SGs by binding RNAs, and following ATP hydrolysis, remodels RNPs in a manner that limits SG assembly (Fig. 7 A). Consistent with this interpretation, we observed an increase in RNA (Fig. 3), G3BP1, and various other RBPs in SG in DDX6 KO cells (Fig. 6, H and I), which could contribute to increased SG formation forming additional intermolecular protein–protein or RNA–RNA interactions (Fig. 2).

Interestingly, the requirement for ATP hydrolysis could explain cell type–specific differences in SG (Majerciak et al., 2023) or PB regulation. In an ATPase active form, DDX6 limits SGs, whereas the inactive form could shift to a scaffolding activity. Further, this could lead to differences in RNP granule numbers

and composition, as seen for PBs in U-2 OS and HeLa cells (Figs. 1 and 2) (Majerciak et al., 2023; Ayache et al., 2015), which express lower and higher levels of DDX6, respectively (Beck et al., 2011; Itzhak et al., 2016). This could also explain differences in protein partition, such as TIA-1 localization in PBs (Fig. S1 A) or being excluded from PBs (Hubstenberger et al., 2017; Kedersha et al., 2005), while eIF4G1 was only seen in SGs (Youn et al., 2018; Kedersha et al., 2005), but can also be detected in PBs (Fig. 1 C) (Hubstenberger et al., 2017). Therefore, depending on its ATPase active or inactive conformation, which could be regulated by expression levels of DDX6 or ATPase activators such as CNOT1 or other interacting proteins, DDX6 is expected to have multiple roles in RNP granule biology (Hondele et al., 2019).

An interesting question is whether a role for DDX6 in limiting SGs could explain any of the pathogenic mutations in DDX6. SGs contribute to tumor progression, and SG-enriched proteins are associated with protein aggregation–related degenerative diseases (Anderson et al., 2015; Shukla and Parker, 2016; Taylor et al., 2016). Moreover, rare mutations in DDX6 cause intellectual disability (Balak et al., 2019). These mutations are known to limit DDX6 interaction with PB proteins and reduce PB formation. However, many of these mutations also lie in the RNA binding interface and therefore may affect DDX6's role in limiting SG formation. Moreover, it is striking that mutations in other DEAD box RNA helicases, such as DHX30 (Lessel et al., 2017), DDX3X (Lennox et al., 2020), or DDX59 (Shamseldin et al., 2013), all cause intellectual disability or other neurodevelopmental disorders. Consistent with a model whereby aberrant RNP granules contribute to neurodevelopmental defects, both DHX30 and DDX3X mutations were shown to form constitutive SGs or SG-like aggregates (Lessel et al., 2017; Lennox et al., 2020).

A second contribution of this work is to identify PB components that can either promote or limit PB–SG docking, which fundamentally occurs through protein–protein, protein–RNA, or RNA–RNA interactions between PBs and SGs. For example, during a stress response in DDX6 KO cells or cells with knockdown of DDX6 and 4E-T, we observed an increase in smaller PBs or PB-like puncta that are predominantly docked to the periphery of SGs (Figs. 4 and 5). The increase in PB number, although minimal, and docking to SGs is also seen upon DCP1A knockdown. One possibility is that DDX6 limits protein interactions between components of SGs and PBs. A role for DDX6 and DDX6 interacting partners in limiting the docking of PB and SG is also consistent with work showing increases in PB–SG interactions in the absence of DDX6 (Majerciak et al., 2023). In principle, this could be achieved directly via DDX6 ATPase activity breaking apart protein interactions or indirectly through the formation of PBs, enriching proteins inside the granule, thereby reducing interaction sites for docking mediating proteins. In the latter model, one anticipates that upon loss of key

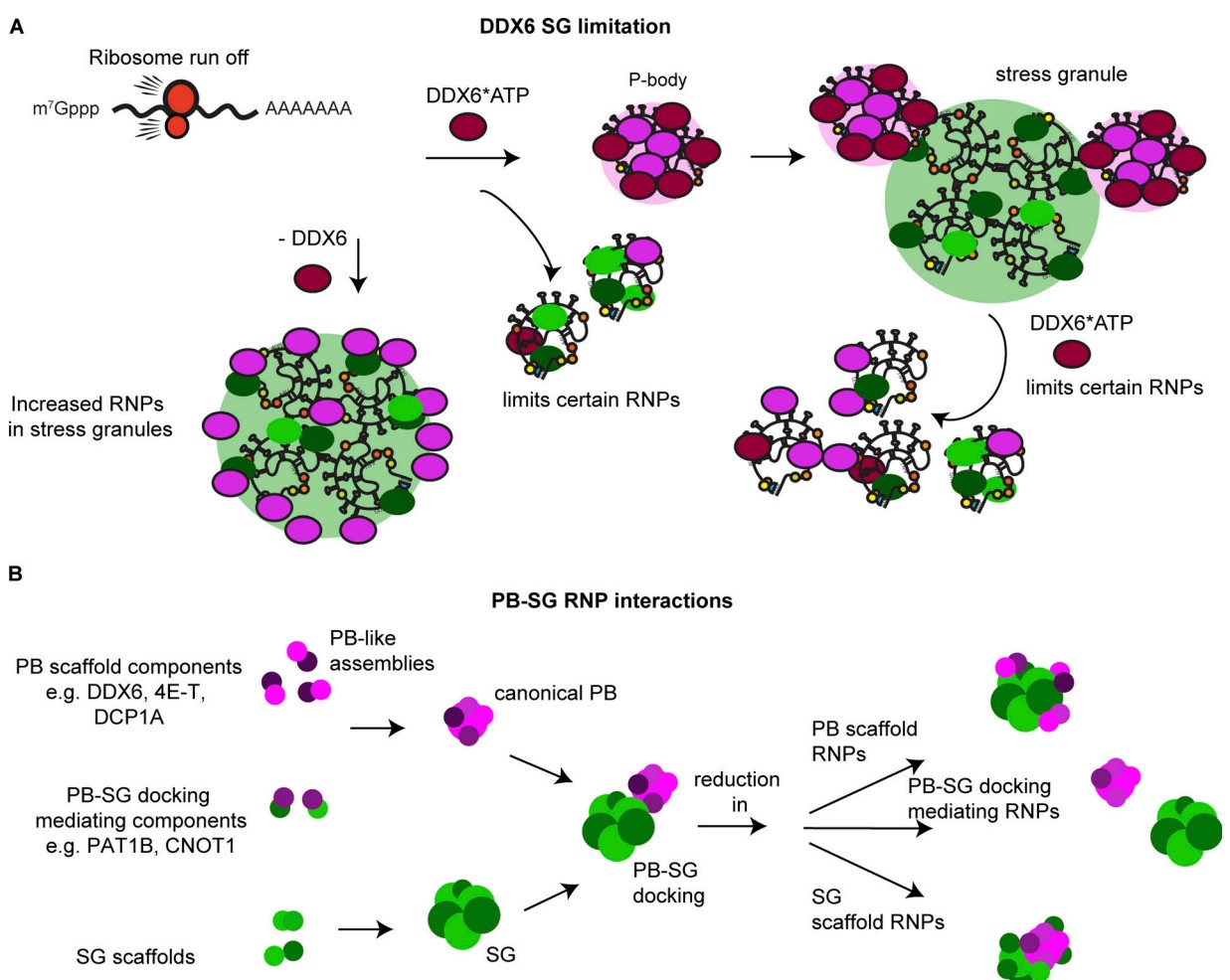

Figure 7. **Model for DDX6 limiting SGs and modulating PB–SG interactions. (A)** After the release of ribosome-free RNA upon stress, in an RNA binding and ATPase-dependent manner, DDX6 forms PBs by binding RNAs, and following ATP hydrolysis, thereby remodeling PB RNPs and preventing some RNP partitioning into PBs. When PB formation reaches steady-state levels and stress granules start to form, using the same mechanism, DDX6 remodels RNPs and prevents them from partitioning into SG. Upon loss of DDX6, SGs are increased due to mislocalization of PB proteins and increased enrichment of RNPs. **(B)** Specific PB RNPs such as DDX6, 4E-T, and DCP1A can act as PB scaffolds and regulate PB assembly and growth while others mediate PB–SG docking, e.g., CNOT1 and PAT1B. Loss of PB scaffolds leads to smaller PBs that increasingly dock to SGs. Loss of PB–SG mediating RNPs abolishes docking. This suggests that a similar principle might apply for SG assembly.

interactions within PBs, more docking-mediating proteins are exposed either due to the loss of specific intra-P-body interactions or because of the increased surface-to-volume ratio on smaller PB-like assemblies (Fig. 7 B). It is also possible that DDX6 could limit PBs and SGs docking by preventing intermolecular RNA–RNA interactions between mRNPs in PBs and those in SGs. This possibility is suggested by the observation that inhibiting the RNA helicase activity of eIF4A increases the docking of PBs and SGs, which implies a role for RNA–RNA interactions in PB–SG interactions (Tauber et al., 2020a). It is notable that other components of PBs enhance docking, with the PB–SG docking reduced by CNOT1 or PAT1B knockdown (Fig. 5, A and B). CNOT1/PAT1B might play a role in linking PBs and SGs through PAT1B interactions with the RNA decapping factors DCP2-DCP1A in PBs and CNOT1 (Ozgur et al., 2010), which is more predominantly seen in SGs (Fig. S1 D). Interestingly, over-expression of the RNA binding proteins, TPP or CPEB1 resembles the DDX6 and/or 4E-T knockdown, leading to smaller PBs or

even smaller PB-like assemblies that now preferentially dock to SGs (Stoecklin and Kedersha, 2013; Wilczynska et al., 2005). Developing an understanding of the specific interactions that modulate PB–SG docking should reveal features of RNP granule surfaces and potentially provide insight into mechanisms of exchange between RNP granules.

We, and others, have shown that PBs and SGs have overlapping as well as unique components (Figs. 1 and S1), indicating the specificity of RNP granule assembly or the presence of a regulatory mechanism that can segregate these components. Our data show that loss of DDX6 leads to increased relocalization of various RBPs and PB components into SGs or docking to SGs (Fig. 4; Fig. 6, H and I; and Fig. S5 G). In fact, DDX6 was shown to bind some SG-enriched RBPs via immunoprecipitation in an RNA-dependent as well as -independent manner (Bish et al., 2015; Ayache et al., 2015). Since we observe DDX6 can limit the partitioning of certain RBPs such as G3BP1 and TIA into SGs (Fig. 6, H and I), we suggest that DDX6 could regulate or limit the

partitioning of these RNPs into granules through remodeling RNP interactions (Fig. 7 A). PB components that can localize in SGs in wild-type cells, such as LSM4, LSM14A, eIF4E, eIF4G1, CNOT1 (Fig. 1 C and Fig. S1 D), or GW182 (Majerciak et al., 2023) show a stronger relocalization into SGs in DDX6 KO cells (Fig. 4 C and Fig. S3 D) (Majerciak et al., 2023). Components that are only in PBs such as DCP1A, EDC4, or 4E-T (Fig. 1, E and F) do not fully relocalize into SGs and form smaller PB-like subassemblies within SGs or at their surface (Fig. 4, A, B, and D–F), resembling docking interactions. This is due to the presence of competing interaction networks (Sanders et al., 2020). If PB components form heterotypic interactions with SG components, they can localize to both granules and easily relocate upon loss of one or the other granule. If PB components prefer to form homotypic interactions, they will not mix within SGs and form distinct assemblies (Ripin and Parker, 2023). Overall, this indicates that most RBPs have the tendency to enrich in both RNP granules most likely due to the localization of their RNA targets, contribute to the corresponding granule maturation through increased multivalency or nodes (Sanders et al., 2020), and that their condensation and segregation into PBs or SGs is a controlled mechanism and impaired upon DDX6 loss.

## Materials and methods

### Cell's growth conditions
Human osteosarcoma U-2 OS cells were maintained in DMEM with 10% FBS and 1% penicillin/streptomycin at 37°C, 5% $CO_2$.

### Cell lines and plasmids
Wild-type and G3BP1/2 KO U-2 OS are kindly provided by Paul Anderson's Lab at Brigham and Women's Hospital, Boston, MA, USA (Kedersha et al., 2016). DDX6 KO using wild-type and G3BP1/2 KO U-2 OS cell lines were generated as follows: The CRISPR/Cas9 guide RNAs targeting two regions within the DDX6 locus were designed using the Integrated DNA Technologies (IDT) CRISPR guide target design tool. Overlapping oligos (DDX6 sgRNA 1, 2 sense and DDX6 sgRNA 1,2 antisense [Table S1]) were annealed in T4 DNA ligase buffer (B0202S; NEB) and ligated into the BbsI-HF (R3539S; NEB) sites in pSpCas9(BB)-2A-GFP (px458) (48138; Addgene) using T4 DNA ligase (B0202S; NEB). To generate DDX6 knockout in U-2 OS and G3BP1/2 KO U-2 OS lines, cells (T-25 flask; 60% confluent) were cotransfected with 3 μg pSpCas9(BB)-2A-GFP- DHX6 sgRNA1+2 and 400-ng of pcDNA3.1-puro using 15 μl of Lipofectamine 2000 (11668019; Thermo Fisher Scientific) according to the manufacturer's instructions. 24 h after transfection, Cas9-GFP expression was observed via fluorescent microscopy. The medium was replaced with a medium containing 2 μg/ml of puromycin (P8833; Sigma-Aldrich). The selective medium was replaced 2 days after transfection. 5 days after transfection, the selective growth medium was replaced with a normal growth medium. When cells reached 80% confluency, cells were serial diluted and plated on 15-cm dishes. Individual colonies were isolated, propagated, and screened via immunoblot analysis.

GFP-tagged DDX6 wild-type and mutant sequences (Table S1) were synthesized and cloned into pTwist CMV by Twist

Bioscience. The generation of the pLenti-ef1-blast-DDX6 lentiviral plasmids was performed as described in Burke et al. (2022). Briefly, untagged DDX6 sequences were amplified via PCR (Table S1), and the sequences were inserted into the XhoI/XbaI sites of pLenti–EF1-BLAST vector using In-Fusion seamless cloning (Takara Bio).

The GFP-labeled G3BP1 cell lines for FRAP and the DDX6 rescue U-2 OS cell lines were generated using a lentivirus system. HEK293T cells (T-25 flask; 80% confluent) were cotransfected with 2.7-μg pLenti-EF1-DDX6-blast or pLenti-EF1-eGFP-G3BP1-blast (Burke et al., 2020), 870-ng of pVSV-G, 725-ng of pRSV-Rev, and 1.4-μg of pMDLg-pRRE (Burke et al., 2020), using 20-μl of lipofectamine 2000. The medium was collected 48 h after transfection and filter-sterilized with a 0.45-μm filter. Then, U-2 OS DDX6 KO cells (T-25 flask; 80% confluent) were transduced with 1 ml of lentiviral stocks containing 10 μg/ml of polybrene (TR-1003-G; Millipore Sigma) for 1 h. DMEM was then added to the flask. 24 h after transduction, cells were reseeded in a T-75 flask containing 10 μg/ml blasticidin (A11139-03; Thermo Fisher Scientific) selective medium. Cells were maintained in selective medium for 4–5 days before returning to normal DMEM.

### Stress conditions and drug treatments
To induce SGs, arsenite (S7400; Sigma-Aldrich; 500 μM), sorbitol (S1876; Sigma-Aldrich; 0.5 M), or hippuristanol (1 μM [Bordeleau et al., 2006]) was added, and the cells were incubated at 37°C for 1 h. To allow for faster stress recovery, cells were incubated with 100 μM arsenite for 1 h. For ribopuromycinylation assays, cells were incubated with puromycin (P8833; Sigma-Aldrich; 10 mg/ml) at 37°C for 5 min prior to fixation.

### Immunoblotting
Cells were washed with ice-cold phosphate-buffered saline (PBS) and lysed with Pierce RIPA Buffer (89900; Thermo Fisher Scientific), 1 mM DTT (R0861; Thermo Fisher Scientific), and 1* cOmplete Mini EDTA-free Protease Inhibitor Cocktail (11836170001; Sigma-Aldrich). Cells were lysed on ice for 20 min (vortexed every few minutes) and then clarified by centrifugation (13,000 RPM for 10 min). 4× Nu-PAGE sample buffer (NP0007; Thermo Fisher Scientific) was added to lysates to a final concentration of 1×, samples were boiled for 5 min at 95°C, and then loaded into 4–12% Bis-Tris Nu-PAGE gel and transferred to a nitrocellulose membrane. Membranes were blocked with 5% nonfat-dried milk in Tris-buffered saline with 0.1% Tween-20 (P9416; Sigma-Aldrich) (TBST) for 1 h and then incubated with primary antibody 1 h at room temperature or overnight at 4°C. Antibody dilutions are listed in Table S1. Membranes were washed 3× with TBST and then incubated with secondary antibody at room temperature for 1 h in 5% nonfat-dried milk in TBST. Membranes were washed 3× again in TBST, and antibody detection was achieved by rocking membranes in Pierce ECL Western blotting substrate (32106; Thermo Fisher Scientific) for 5 min.

### siRNA-mediated knockdown
For siRNA knockdowns, 200,000 cells per well were seeded into six-well plates. After 24 h, cells were transfected with 40 nM

ON-TARGETplus siRNA SMARTpool (Horizon Discovery) using Lipofectamine RNAiMAX (13778-150; Invitrogen). Per reaction, 5 µl lipofectamine was added to 250 µl Opti-MEM Medium. 20 nM siRNA was added to another tube with 250 µl Opti-MEM Medium. Both were combined, vortexed, and incubated for 20 min at RT. 500 µl siRNA- lipofectamine mix was added per well with 2 ml media. 24 h after transfection, cells were trypsinized and 200 µl (around 55,000 cells) were seeded onto a glass coverslip in 24-well plates to be fixed for imaging or lysed for Western blots the day after.

### Immunofluorescence (IF) and single molecule RNA FISH
55,000–65,000 cells were seeded on a glass coverslip in 24-well plates. The next day, untreated or stressed or drug-treated cells were fixed with 4% paraformaldehyde in PBS for 10 min and permeabilized with 0.1% Triton X-100 (AC327371000; Thermo Fisher Scientific) in RNase-free water for 5 min at room temperature. Coverslips were blocked with 5% BSA (126593; Millipore Sigma) in PBS for 1 h. Primary antibody incubation in 5% BSA in PBS was performed for 1 h at RT followed by three PBS washes. Secondary antibody (coupled to Alexa Fluor Dyes) incubation in PBS was performed for 1 h at RT, followed by three PBS washes. All antibodies used in this study were from commercial sources. Antibodies and dilutions are listed in Table S1. For IF, coverslips were mounted with Prolong Glass Antifade Mountant with NucBlue Stain (P36981; Thermo Fisher Scientific).

smFISH was performed as described in the manufacturer's protocol (Stellaris RNA FISH protocols | LGC Biosearch Technologies). RNA FISH probes are described in Khong et al. (2017). Coverslips were mounted with VECTASHIELD Antifade Mounting Medium with DAPI (101098-044; VWR) and sealed with nail polish.

### Imaging
Imaging was performed at room temperature using the inverted Nikon Ti Eclipse spinning disk confocal microscope with a 100× NA 1.4 oil immersion objective and a 2× Andor Ultra 888 EMCCD camera (BioFrontiers Advanced Light Microscopy Core) (Figs. 1, S1, S2, S3, and 4), the Nikon Spinning Disk Super Resolution by Optical Pixel Reassignment (SoRa) microscope with a 20× NA 0.75 air (Fig. S2 H), 100× NA 1.45 oil immersion objective, and a Hamamatsu ORCA Fusion BT sCMOS Camera using 1× (Figs. 3, 4, S4, 5, S5, and 6) or 2.8× magnification (Fig. 4 F). Nikon NIS-Elements was used as acquisition software. Every experiment was performed in biological replicates/different days and indicated in the figures. More than three images were recorded at room temperature per biological replicate. The number of analyzed cells is indicated in the figures.

### FRAP
FRAP assays were performed using an inverted Nikon A1R laser scanning confocal microscope equipped with an environmental chamber, a 100× NA 1.5 oil objective, and Nikon Elements software. The mean intensity of the prebleach and bleached region was quantified using Nikon Elements software, and recovery intensities were normalized to the mean of the pre-bleach intensities. The curve for exponential recovery, mobile fraction, and $T_{1/2}$ was determined using the Excel Solver plugin.

### Image analysis and quantification
Image processing was conducted using the Fiji image-processing package (http://fiji.sc/Fiji), with all shown images in the manuscript being the maximum intensity projection of a series of z-slices. Minimum and maximum display values were set in ImageJ for each channel. The fluorescence intensity of various RBPs in the cytoplasm, and SGs versus PBs was determined using the Fiji image-processing package by manually selecting random areas (Fig. 1, Fig. S1; Fig. 6, H and I; and Fig. S5, G and H) or indicated lines (Fig. S1 D) within the cell.

Quantification of fluorescence intensities (including DDX6 SG partition coefficients), SG and PB counts, volumes, areas, EDC4 spots, and smFISH spots was performed on either the 3D image or a single plane using the spot, surface, and cell functions from Imaris Image Analysis Software (Bitplane) (University of Colorado-Boulder, BioFrontiers Advanced Light Microscopy Core) as described before (Khong et al., 2018). Partition coefficients were determined by the mean fluorescence intensity in granule/cytoplasm. In the PB quantification of the DDX6 rescue experiments, cells without any PB rescue (no transduction) were excluded. The collective SG+PB areas in Fig. S2 G were determined using the Imaris channel arithmetics tool by merging both channels (Fig. S2 G, middle) and creating new SG+PB surfaces on the merged channel (Fig. S2 G, right). New surfaces were filtered on an SG fluorescence median intensity threshold to select SG with docking PBs.

Replicate values were normalized to control conditions. Plots were created using R-studio and the ggplot package.

### Statistical analysis
Statistical analysis on per-cell quantification was performed using R-studio and the stat_compare_means function. For measurements per cell (area, granule number, and RNA spots) the unpaired, two-sided $t$ test on individual cell data points (from two to three biological replicates, each replicate shown in red-orange-pink and the mean replicate values are indicated as triangles) was used and each perturbation compared with the wild-type or control samples. For the time courses and partition coefficient, means were calculated in Excel and an unpaired, two-tailed $t$ test was used.

### Online supplemental material
Fig. S1 shows PBs grow in number before the formation of SGs, share SG proteins but also have unique proteins. Fig. S2 addresses DDX6 KO validation and supporting data for DDX6 limiting SG assembly. Fig. S3 describes supporting data for increased RNA enrichment and the formation of smaller PB-like assemblies in DDX6 KO cells. Fig. S4 provides supporting data for siRNA knockdowns of various PB components that lead to an increase in SGs, docking of smaller PBs, or loss of PB–SG docking. Fig. S5 shows cell line validations, and supporting data for DDX6 mutants shows different PB phenotypes but no effects on SGs. Table S1 (excel spreadsheet with 5 sheets) lists the DNA oligo sequences for cloning including gRNAs, pTwist DDX6 sequences, siRNAs, antibodies, and used dilutions for IF or Westerns.

**Data availability**

All raw imaging data are available upon request.

## Acknowledgments

We thank Theresa Nahreini (Cell Culture Facility, Department of Biochemistry, University of Colorado, Boulder, CO, USA) and Dr. Joseph Dragavon and Dr. Jian Wei Tay (BioFrontiers Institute Advanced Light Microscopy Core Facility, University of Colorado, Boulder, CO, USA; RRID: SCR_018302). Spinning disc confocal microscopy was performed on a Nikon Ti-E microscope supported by the BioFrontiers Institute, University of Colorado, Boulder, CO, USA, and data analysis was performed using the Analysis Workstation and the software package Imaris (supported by NIH 1S10RR026680-01A1). Hippuristanol solution was kindly gifted by Prof. Jerry Peletier (McGill University, Montreal, Quebec, Canada).

The research reported in this publication was supported by Howard Hughes Medical Institute (HHMI) funds (R. Parker). N. Ripin is a National Institutes of Health (NIH) K99 awardee (K99GM148758).

Author contributions: N. Ripin and R. Parker conceived the project. N. Ripin, L. Macedo de Vasconcelos, and D.A. Ugay performed experiments. N. Ripin analyzed the data. N. Ripin and R. Parker wrote the manuscript.

Disclosures: The authors declare no competing interests exist.

Submitted: 5 June 2023

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

# Supplemental material

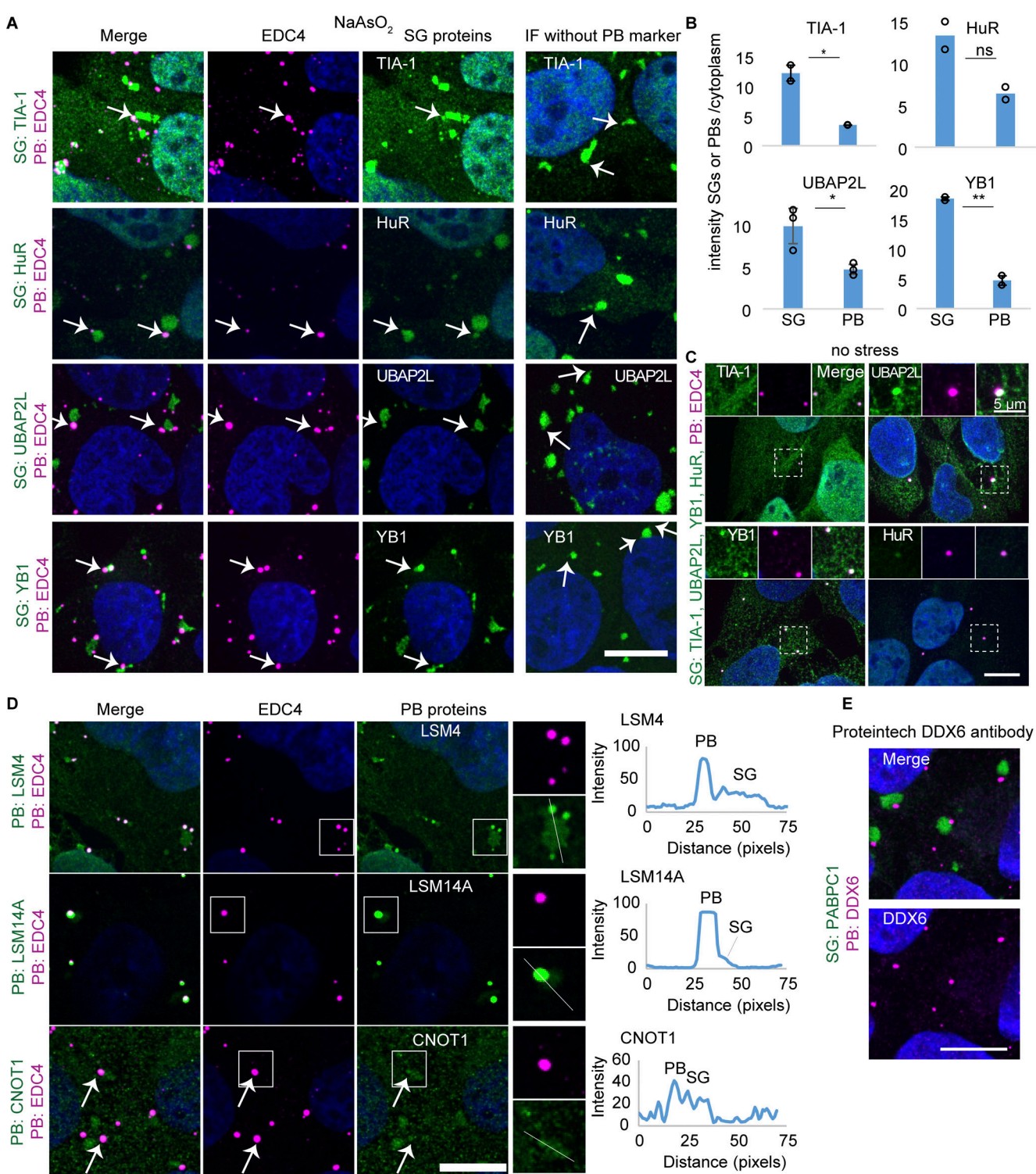

Figure S1.   **Temporal changes in PB and SG formation and their composition. (A)** IF images of SG proteins (green) that also colocalize with PBs (EDC4 IF, magenta). Panel "IF without PB marker" is a bleed-through control, highlighting visible PB localization even without co-staining for a PB marker. Arrows point to examples. **(B)** Quantification of mean intensity granule/cytoplasm in SGs and PBs of the same proteins as in A. Each data point represents the mean value of one replicate. Error bars represent the standard deviation of three (UBAP2L, YB1) or two (HuR, TIA-1) independent replicates. **(C)** IF images of SG proteins (green) that colocalize with PBs (EDC4 IF, magenta) without stress. **(D)** Left: IF images of PB proteins (LSM4, LSM14A, CNOT1 IF, green, EDC4 IF, magenta) that also colocalize in SGs (no marker). Weak CNOT1 PB localization is indicated by arrows. Right: Line plots for PB and SG intensity of proteins shown on the left and corresponding lines indicated in inlets. **(E)** IF image of DDX6 using a second antibody (Proteintech). IF images are representative images of three independent biological replicates with more than three images analyzed per replicate. Scale bar, 10 µm if not indicated. ***P < 0.001, **P < 0.01, *P < 0.05 (unpaired, two-tailed t test).

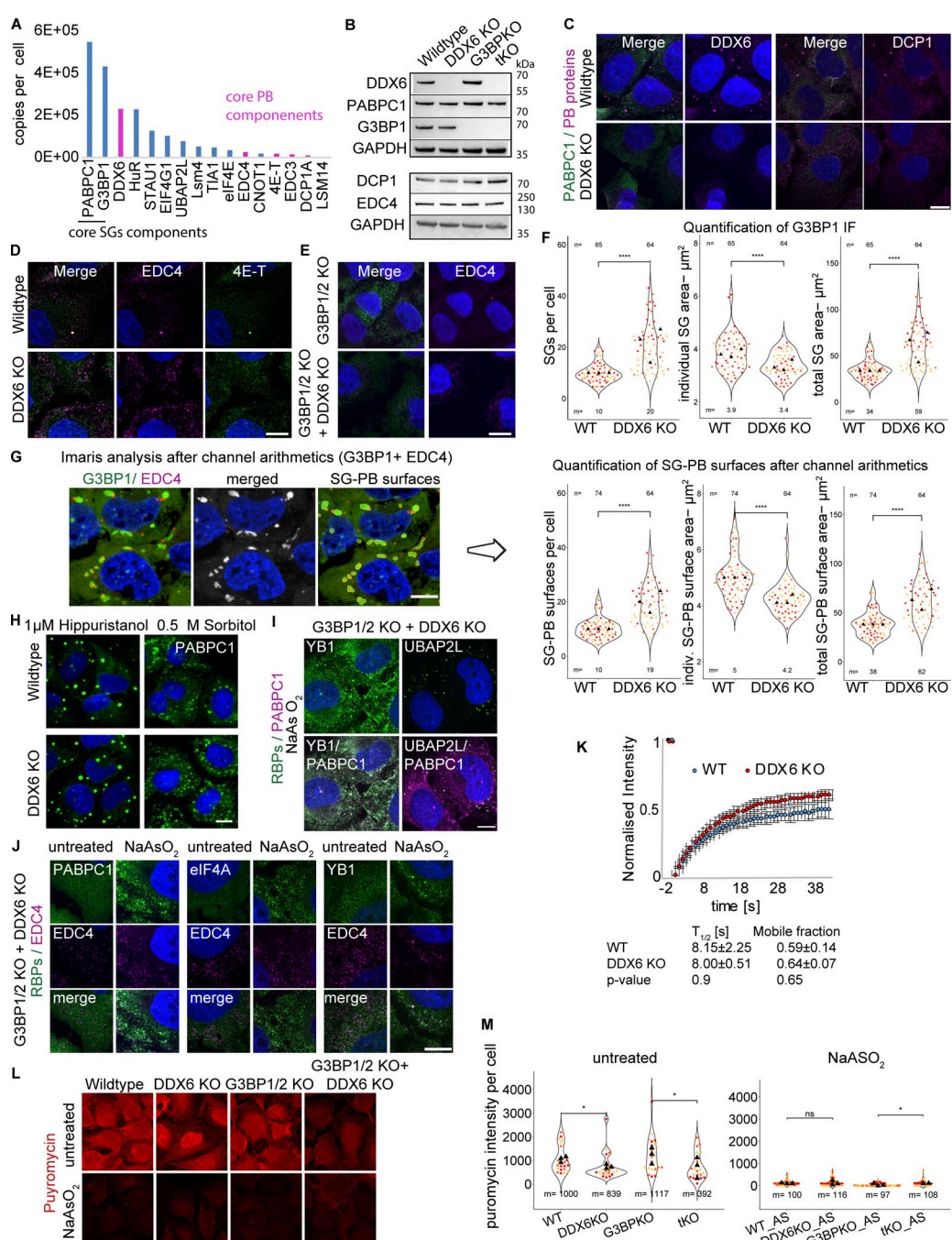

Figure S2. **DDX6 KO validation and supporting data for DDX6 limiting SG assembly. (A)** Bar plot of a few SG- and PB-associated protein levels (derived from the U-2 OS proteome [Beck et al., 2011]) with RNP granule localization patterns indicated. **(B)** Immunoblot of the main four cell lines used in this study, depicting DDX6, PABPC1, G3BP1 (top), DCP1A, EDC4 (bottom), and GAPDH as a loading control. G3BP1/2 KO and G3BP1/2+DDX6 triple KO cells are abbreviated with G3BPKO and tKO, respectively. **(C)** IF images of non-stressed conditions showing loss of DDX6 (left, magenta) and PBs upon DDX6 KO and diffuse or punctate signal for DCP1A (right, magenta). **(D)** As in C but for EDC4 and 4E-T. **(E)** IF images of non-stressed G3BP1/2 KO and G3BP1/2KO+DDX6KO cells showing loss of PBs upon DDX6 KO and diffuse or punctate signal for EDC4 (right, magenta). **(F)** As in Fig. 2, C–E but using G3BP1 IF images: Quantification of SG number, mean SG area, the total SG area per cell. **(G)** G3BP1 IF Images were used for Imaris channel arithmetics (merging the SG and PB channel) to determine the combined SG+PB surface parameters. **(H)** IF images (20× magnification) showing the increase in SGs upon DDX6 KO (PABPC1 IF, green) upon Hippuristanol and Sorbitol stress. **(I)** IF images of YB1 and UBAP2L (green) in stressed G3BP1/2+DDX6 triple KO cells. **(J)** IF images of PABPC1, eIF4A, and TIA-1 (green) colocalized with EDC4 (magenta) in stressed G3BP1/2+DDX6 triple KO cells. **(K)** Quantification of GFP-G3BP1 FRAP in SGs (0.5 mM arsenite) in wild-type and DDX6 KO cells showing no significant differences ($t$ test) in the recovery or mobile phase. Error bars represent standard deviation of three independent biological replicates. **(L)** IF images of ribopuromycinylation assays. **(M)** Quantification of the mean puromycin intensity per cell. For quantification, data points from three biological replicates are shown in red-orange-pink, the mean replicate values are indicated as triangles, and mean values are shown on the bottom of the image (m). ****$P \leq 0.0001$, ***$P \leq 0.001$, **$P \leq 0.01$, *$P \leq 0.05$, ns $P > 0.05$ (unpaired, two-tailed $t$ test on individual data points). IF images are representative images of three independent biological replicates with more than three images analyzed per replicate. Scale bar, 10 μm if not indicated. Source data are available for this figure: SourceData FS2.

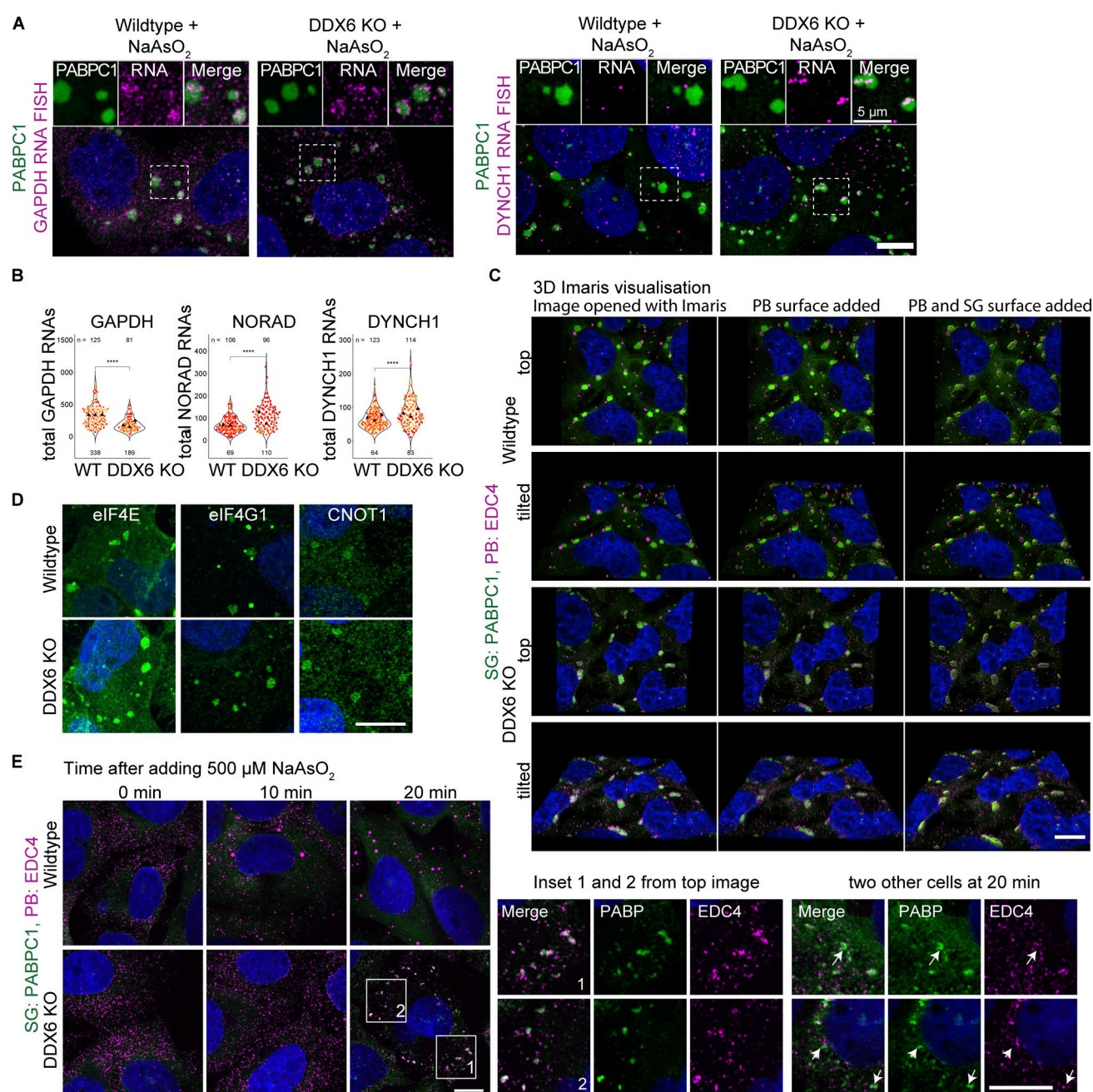

Figure S3. **Supporting data for increased RNA enrichment and the formation of smaller PB like assemblies in DDX6 KO cells. (A)** IF images comparing GAPDH and DYNCH1 RNA localization in SGs in wild-type and DDX6 KO cells. **(B)**. Quantification of the total number of GAPDH, DYNCH1, and NORAD RNAs in the cytoplasm. For quantification, data points from three biological replicates (two for NORAD) are shown in red-orange-pink, the mean replicate values are indicated as triangles, and mean values are shown at the bottom of the image (m). ****P ≤ 0.0001, ***P ≤ 0.001, **P ≤ 0.01, *P ≤ 0.05, ns P > 0.05 (unpaired, two-tailed *t* test on individual data points). **(C)** Imaris 3D visualization in wild-type (top) and DDX6 KO (bottom) cells. Z-stacks are opened with Imaris (left) and PB surfaces (center) or PB and SG surfaces (right) are added. **(D)** IF images showing eIF4E, eIF4G, and CNOT1 localize to both granules in wild-type cells or in SGs in DDX6 KO cells. **(E)** Top: IF images showing the increase in SGs and PBs/PB-like assemblies (EDC4 IF, magenta; PABPC1, green) in wild-type and DDX6 KO cells at 10 and 20 min of stress treatment, to highlight the formation of interacting PB-like assemblies and smaller SGs in DDX6 KO cells. EDC4 IF signal is overamplified to highlight the lack of PB-like assemblies at earlier timepoints and in cells without SGs. White boxes denote the insets at the bottom. Bottom: Close-up view of the two indicated insets and two additional close-up views at 20 min of other cells with less or smaller PB-like assemblies.

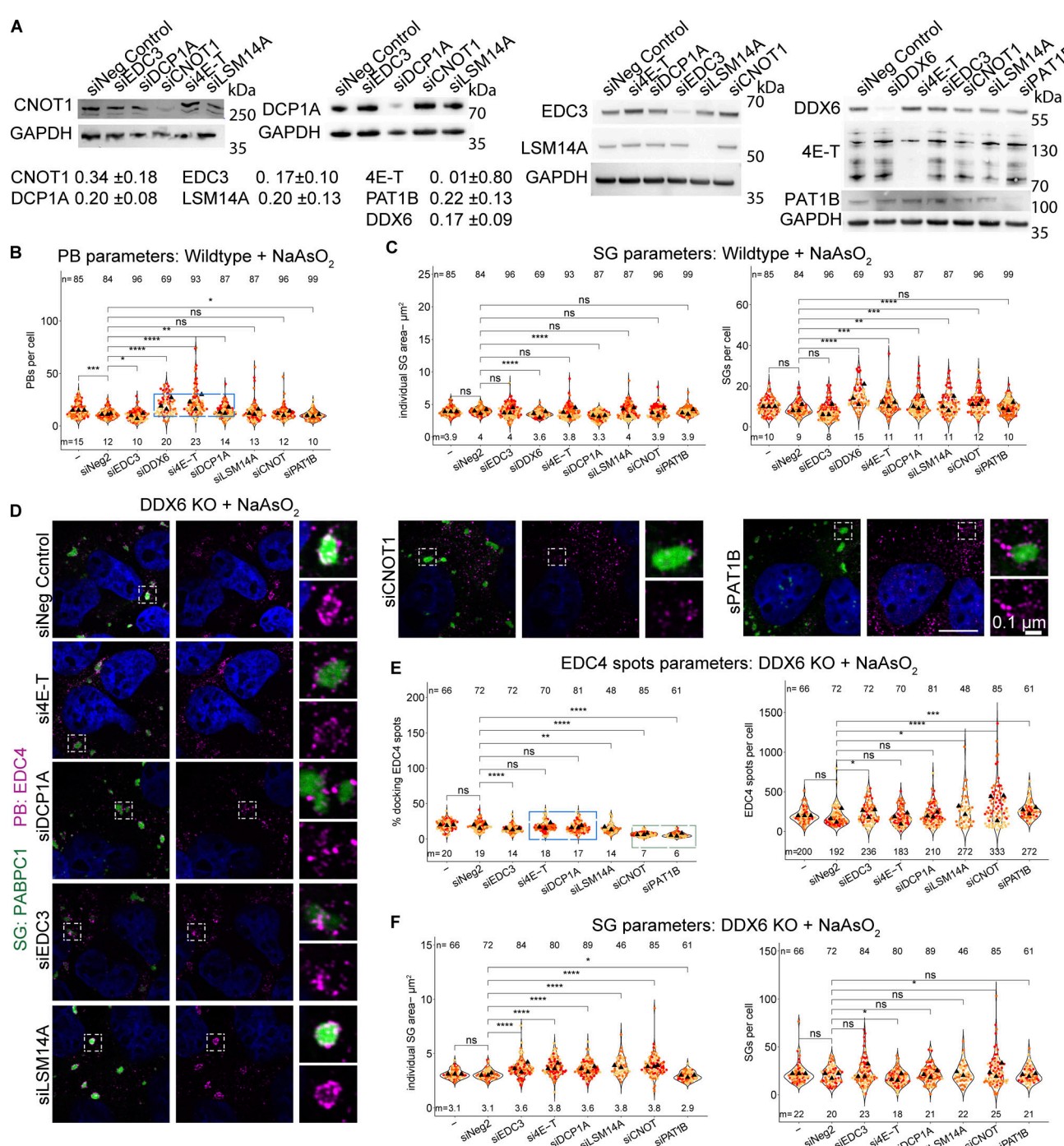

Figure S4. **Supporting data for siRNA knockdowns of various PB components that lead to an increase in SGs, docking of smaller PBs, or loss of PB–SG docking. (A)** Example immunoblots for the corresponding proteins after siRNA knockdowns. Fraction of the remaining protein is indicated (normalized to the corresponding GAPDH control) with errors representing the standard deviation of three biological replicates. **(B)** Quantification of other PB parameters related to Fig. 5 B. The blue box highlights conditions that lead to an increased number of PBs. **(C)** Quantification of other SG parameters related to Fig. 5. **(D)** IF images of EDC puncta (IF, magenta) and SGs (PABPC1, green) in DDX6 KO cells upon siRNA knockdown of PB proteins as in Fig. 5. **(E)** Quantification of the number of EDC4 spots per cell (right) and % docking interactions with SGs (left). The blue box highlights conditions that, in wild-type cells, lead to an increase in smaller PBs but show no effect in DDX6 KO cells. The green box highlights conditions that lead to reduced PB–SG docking. **(G)** Quantification of other SG parameters related to Fig. 5 D. All IF images are representative images of three independent biological replicates with more than three images analyzed per replicate. For quantification, data points from three biological replicates are shown in red-orange-pink, the mean replicate values are indicated as triangles, and mean values are shown at the bottom of the image (m). The number (n) of analyzed cells is shown at the top of the image. ****P ≤ 0.0001, ***P ≤ 0.001, **P ≤ 0.01, *P ≤ 0.05, n.s. P > 0.05 (unpaired, two-tailed $t$ test on individual data points). Scale bar, 10 μm if not indicated. The Non-Targeting Control siRNA Pool #2 was abbreviated with siNeg Control or siNeg2. Columns marked as "–" are untransfected cells. Source data are available for this figure: SourceData FS4.

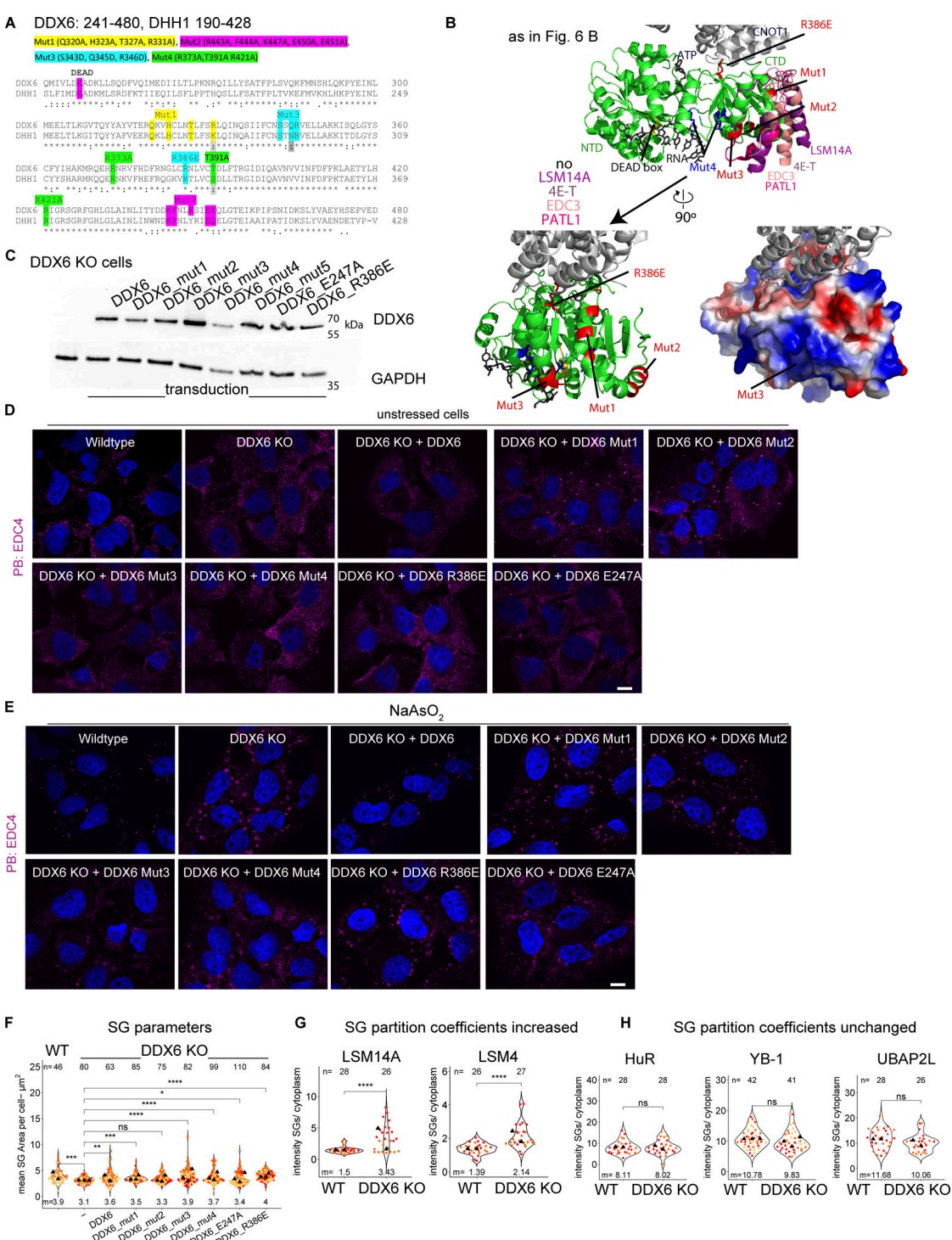

Figure S5. **Cell line validations and supporting data for DDX6 mutants show different PB phenotypes but no effects on stress granules. (A)** DDX6 and the yeast ortholog DHH1 amino acid sequence alignment using Multiple Sequence Alignment—CLUSTALW. Amino acids 241–480 and 190–428 for DDX6 and DHH1, respectively, are shown and mutations indicated. **(B)** Top: As in Fig. 6 B. Bottom left: DDX6 structure as on top but without the binding partners and turned by 90° to highlight Mut1-3. Bottom right: Surface electrostatic map shows that Mut3 is located within a positive patch (blue color), indicating that Mut3 might be an inactive RNA binding mutant. **(C)** Immunoblot of stably transduced cell lines (Mut5 not used in this study). **(D)** IF images of unstressed wild-type, DDX6KO, and rescue cell lines showing PB (EDC4 IF, magenta) phenotypes. **(E)** IF images of stressed wild-type, DDX6KO, and rescue cell lines showing PB (EDC4 IF, magenta) phenotypes. **(F)** Quantification of mean SG areas. **(G and H)** Quantification of mean protein intensity in SG/cytoplasm in wild-type and DDX6 KO cells. All IF images are representative images of three independent biological replicates with more than three images analyzed per replicate. For quantification in B–E, data points from three biological replicates are shown in red-orange-pink, the mean replicate values are indicated as triangles, and mean values are shown at the bottom of the image (m). The number (n) of analyzed cells is shown at the top of the image. ****P ≤ 0.0001, ***P ≤ 0.001, **P ≤ 0.01, *P ≤ 0.05, n.s. P > 0.05 (unpaired, two-tailed t test on individual data points). Scale bar, 10 μm if not indicated. Columns marked as "−" are untransduced cells. Source data are available for this figure: SourceData FS5.

**Provided online is Table S1. Table S1 that lists the DNA oligo sequences for cloning including gRNAs, pTwist DDX6 sequences, siRNAs, antibodies, and used dilutions for IF or Westerns.**

