## [Peer Review File · The Journal of Cell Biology]

DDX6 modulates P-body and stress granule assembly, composition and docking

Nina Ripin, Luisa Macedo de Vasconcelos, Daniella Ugay, and Roy Parker

Corresponding Author(s): Roy Parker, University of Colorado Boulder

Review Timeline:

Submission Date:	2023-06-05
Editorial Decision:	2023-07-26
Revision Received:	2023-12-20
Editorial Decision:	2024-02-01
Revision Received:	2024-02-07
Editorial Decision:	2024-02-19
Revision Received:	2024-02-26

Monitoring Editor: Karla Neugebauer

Scientific Editor: Andrea Marat

Transaction Report:

DOI: <https://doi.org/10.1083/jcb.202306022>

July 26, 2023

Re: JCB manuscript #202306022

Dr. Roy Parker
University of Colorado Boulder
Dept of Biochemistry
JSCBB
Boulder, CO 80309

Dear Roy,

Thank you for submitting your manuscript entitled "DDX6 modulates P-body and stress granule assembly, composition and docking" to the Journal of Cell Biology. Thanks also for your patience as it took longer than typical to provide you with a decision on your study. The manuscript has been evaluated by expert reviewers, whose reports are appended below. Unfortunately, after an assessment of the reviewer feedback, our editorial decision is against publication in JCB.

We are concerned that the work is mainly descriptive and lacks both mechanism and novelty, though the issues the experiments expose are topical and important. All three reviewers require improvements to the experiments themselves, which could support the characterization of the role of DDX6 in SG and PB dynamics. The quality of the imaging, including the extent to which molecular components can define SGs and PBs, was criticized. Most concerning are reviewer 1 and 2's comments that several of the introductory figures may reflect previously published experiments from several labs, including the Parker lab. They indicate that the literature has not been completely cited, giving credit to labs that have previously published experiments very closely aligned with these experiments. We hope that this situation will be remedied if you choose to resubmit (including a resolution of the overlaps with the work from Majercki et al. now published as doi: 10.1093/nar/gkad585). Reviewer 2 suggests the possibility that some of the figures overlapping with published studies could be referenced and discussed in the introduction, allowing you to construct a manuscript that focuses on the more novel experiments that support a concise conclusion.

Therefore, while your manuscript is intriguing, I feel that the points raised by the reviewers are more substantial than can be addressed in a typical revision period. If you wish to expedite publication of the current data, it may be best to pursue publication at another journal. Our journal office will transfer your reviews upon request.

Given interest in the topic, I would be open to resubmission to JCB of a significantly revised and extended manuscript that fully addresses the reviewers' concerns including the presentation of new experimental work addressing all of their comments as well as scholarly treatment of the work in the context of previous published studies. If you would like to resubmit this work to JCB, you must contact the journal office to discuss an appeal of this decision. To place an appeal please prepare a detailed response to the reviewers along with a revision plan outlining how you will address all experimental issues. Please note that we may discuss an appeal with the original reviewers. Our goal with the appeal process is to ensure that planned revisions seem appropriate to potentially address all concerns before authors spend time and resources on revisions. Please note that priority and novelty would be reassessed at resubmission, and that a revised study would be subject to further peer-review by the original reviewers or suitable replacements.

Regardless of how you choose to proceed, we hope that the comments below will prove constructive as your work progresses. We would be happy to discuss the reviewer comments further once you've had a chance to consider the points raised in this letter. You can contact the journal office with any questions, cellbio@rockefeller.edu or call (212) 327-8588.

Thank you for thinking of JCB as an appropriate place to publish your work.

Sincerely,

Karla Neugebauer, PhD
Monitoring Editor

Andrea L. Marat, PhD
Senior Scientific Editor

Journal of Cell Biology

Reviewer #1 (Comments to the Authors (Required)):

Ripin et al. investigated the contribution of DDX6 to SG-PB docking, SG composition, and SG and PB formation. This is an important contribution to understanding the interactions between and dynamics of SGs and PBs, stress-responsive membraneless organelles. I am enthusiastic about this work and think it should ultimately be published, however additional work and revision are needed to provide sufficient evidence for the conclusions drawn and to appropriately set this work in the context of the current literature in the field.

Major concerns:

- Nomenclature: it would be helpful to understand the criteria for designating a condensate as a SG or a PB:
 - Since some of the phenotypes contain combinations of proteins that are typically restricted to SGs or PBs (i.e. the SGs in DDX6 KO cells treated with arsenite), it does not seem appropriate to call these SGs or PBs, as they seem to represent some kind of fusion of the two.
- Immunofluorescence images lack sufficient protein markers (or use inappropriate markers) to unambiguously assign them as SGs or PBs (more markers will help address above point). In particular, the following figures should be addressed: Fig 2 A, E, F, G; Fig 4 B, C; Fig 6 C, D; Fig S2 F. Please see details in "additional comments" section below.
- Work needs to be placed in context of additional relevant publications:
 - Sanders et al. Cell 2020:
 - Presents continuum from single phase (i.e. mixed SGs and PBs) to multiphase granules (i.e. separate SGs and PBs) and the role of RNA and protein interactions in determining the spatial arrangement.
 - Stoecklin & Kedersha Adv Exp Meth Biol. 2013 (and/or references contained within):
 - TTP and BRF1 are both found in SGs and PBs and contribute to SG-PB docking/fusion, so DDX6 is not the first protein of this kind, as suggested in the abstract.
 - Several conclusions regarding how DDX6 contributes to SG composition, docking etc. depend on the assumption that the DDX6 mutants share the phenotypes of the analogous DHH1 mutants (ie impaired ATPase activity or RNA binding). The effect of these mutations on DDX6 activity should be validated in human cells, or at the least the DDX6 and DHH1 sequence alignment should be included to show how similar the sequences are and increase the confidence in the conclusions drawn.

Additional comments:

Results:

- loss or restoration of constitutive PBs: U2OS are not the best model for this, as few U2OS cells have any constitutive PBs. Would be better to perform in HeLa or Cos7 cells.
- Fig 2A,E,F: SG quantification based on only one SG marker (PABPC) - should have at least one additional SG marker (G3BP1, eIF3b, or eIF4A).
 - Are these canonical SGs? Based on figure 4, they are not since they contain PB proteins, thus it seems inappropriate to call them SGs.
- In the text, PBs are described as docking to SGs in DDX6 KO cells (lines 204-222, Fig 4A, B). To me, 'docking' suggests two entities making contact at their edges. In WT conditions I agree that the PBs are docking at SGs, as typically observed under arsenite. However, in the case of the DDX6 KO cells 'docking' does not seem to accurately describe the relationship between SGs and PBs, as many PB-like puncta are not at the periphery of the SG but appear to be inside of it. While some nuance to the type of PB-SG association is discussed the description should be modified for clarity. See related comment above about definitions as well.
- Fig 2D - Since the "SGs" in DDX6 KO cells contain PB proteins (per Fig 4), it is conceivable that they are actually a mixture of SG and PB components - therefore, it would be informative to include a count of area covered by SGs and PBs (collectively) in WT cells for comparison. It may be that this will show no difference between WT and DDX6 KO.
- Lines 174-176/ Fig 2E,F : since granules in the DDX6 KO cells contain both PB and SG proteins, the difference in their formation and disassembly rates from WT U2OS SGs could be due to a PB-like composition. From your data (Fig 1B) and that of others, we know that PBs form earlier than SGs in response to stress. Therefore, given that the granules in DDX6 KO cells contain PB proteins, it could be that their rapid formation and delayed disassembly is due to a PB-like composition. Including a PB marker in these experiments would allow for comparison to PB assembly and disassembly rate in WT cells to make this distinction between DDX6 "limiting SG assembly" or resulting in PB-like granules that inherently form more quickly and disassemble more slowly.
- Fig 2G/S2 F - need WT cells and additional SG and PB markers on the same sample to examine co-localization and possible formation of SGs (or SG-like granules) in the triple KO.
- Line 317: 'rescue' of PBs is misleading since few cells have PBs in the absence of stress in U2OS cells. "nucleation" would be more accurate.
- Conclusions from the DDX6 mutant experiments largely depend on the assumption that the mutations in DDX6 have the same effect as in yeast DHH1. The mutations should be validated in order to draw these conclusions. E.g. - does mut 4 impair ATP hydrolysis activity?

- Figure 4 D,C: granules referred to as SGs, but no SG marker shown.
- Figure 6C, D: additional PB markers need to be included to confirm presence or absence of PBs in cells expressing the different DDX6 mutants. It is possible that some DDX6 mutants may not be recruited to PBs, but PBs may still form. Or, DDX6 aggregations may form, but other PB proteins are not recruited, and thus PBs have not formed. contrast should be increased on many images as they are difficult to see (e.g. Fig S2 D, Fig S4 A, Fig S5 C)
- Fig S2 E - bottom right panel - no cells?
- Fig S2 F G1/G2/DDX6 KO arsenite trmt: is it possible these cells are unhealthy and the eIF4a staining reflects how any protein might look? Need non-SG control protein or additional SG protein marker.
- Fig 2G: need WT control (will show arsenite and PABPC antibody behaving as expected)

Methods:

- Table S1 not included, unable to fully assess methods.
- 500 μ M arsenite for 1 hr is a very strong dose. Do these phenotypes occur at lower arsenite doses or in response to other stresses? Or does the system have to be pushed to its max to get these results?
- It is surprising that 1:100 dilution was needed to see DDX6 in SGs. DDX6 localization to SGs and PBs has already been documented.

Miscellaneous minor points:

- Ugay affiliation missing
- Line 157 "cells" "cell"
- Fig 6B and Fig S5A: DAED box DEAD box
- Several places throughout MS where SG or PB should be plural.
- Labeling of extended and supplemental figures is confusing and makes it difficult to find the correct figures. For example, in the text Fig. S3 J and H are referenced. However, Fig. S3 does not have "J" or "H" and is not the correct figure. Rather, "Extended Fig. 3" should be referenced, which is contained in "Fig. S2". Even with the referencing corrected, this remains confusing, as there is no clear distinction within Fig. S2 between Extended Fig. 2 and 3. Please revise.
- Fig 5 is referenced in the text prior to figure 3. Please revise so figures are references in the order they are presented.
- Line 153: Fig 2A referenced for triple KO but that is not part of the figure.
- Are Fig 6B and Fig S5A the same? Please correct/clarify.
- Fig S1 F: labels missing - which protein is each graph for?

Reviewer #2 (Comments to the Authors (Required)):

This study reports that the RNA helicase DDX6, a known P-body (PB) component, can also limit the formation of stress granules (SG) in an ATPase-dependent manner, and that several PB proteins can influence the composition of SGs and their docking with PBs. The experiments show how DDX6, wt or mutant, and a few other proteins affect PB and SG number and size, as well as PB-SG docking. They involve immunostaining of classical PB and SG markers and a few single molecule RNA FISH, in a single cell line (U-2 OS) upon a single stress (arsenite).

At this stage, the manuscript presents well-conducted experiments but of limited novelty. First, that silencing of some PB proteins affects the composition of SGs has been already reported in the literature. Second, the impact of these proteins on SG size was extensively described by Majerciak et al. in a preprint deposited on BioRxiv (2021). Furthermore, it seems difficult to build a strong model, based only on variations of SG size and PB-SG docking, particularly when the effects are significant but quantitatively minor.

Major concerns:

1/ Since the experiments described from lines 97 to 148 essentially replicate data published by the Parker (Buchan et al., 2008, Jain et al., 2016) and other groups (Mollet et al., 2008, Kedersha and Anderson, 2007; Hubstenberger et al., 2017; Kedersha et al., 2005), these results could be all described in the introduction. Reference should be added for the well-known observation that PABPC, G3BP1 and polyA RNA are not enriched in PBs (line 122), and for the fact that DDX6 is more abundant than other PB proteins (line 144) (a Table in Ayache et al., 2015, includes the analysis presented in Fig S2A).

2/ What is the positioning of the authors with respect to the Majerciak et al preprint (BioRxiv 2021). This preprint is entitled "RNA helicase DDX6 in P-bodies is essential for the assembly of stress granules". It addresses the exact same question, with similar approaches. It already describes that the number of arsenite-induced SGs increases after DDX6 silencing, in an ATPase-dependent manner (using the same E247A mutant). Yet, surprisingly, this preprint is cited only once in the result section, for a minor reason (line 386), and not at all in the discussion.

3/ Throughout the manuscript there are ambiguities between organelles, organelle-like foci/assemblies and organelle components, which weaken the conclusions. (i) In G3BP1/2 KO cells, which are devoid of SGs, DDX6 KO leads to the reappearance of "SG-like foci", while the conclusion of the paragraph is about DDX6 limiting the formation of "SGs" (lines 184-186). In view of their completely different morphology, what is the link between "SG-like foci" and SGs beyond the fact they share PABPC1 and eIF4A (Fig 2G and S2F)? (ii) Following the silencing of DDX6, which abolishes PBs, some PB proteins accumulate at the periphery of SGs, in a pattern described as "PB-like assemblies" (lines 205-210), while others localize in both the periphery and the central region of SGs (lines 218-219). Probably by mistake, 4E-T is included in the two classes. More

important, when docking is analyzed, the conclusion is that DDX6 reduces "PB"-SG docking (lines 283-284), despite the absence of PBs in these conditions. It was already reported that EDC4 and DCP1 relocalize in SGs when PBs are abolished using siDDX6 (Mollet et al., 2009; Serman et al., NAR 2007, PMID: 17604308), except that these authors did not observe a stronger accumulation at the SG periphery. Does it depend on the cell line? Or on the efficiency or timing of the silencing?

4/ The study of SG size following silencing of various factors raises several questions. (i) What do we learn from the smFISH experiments (Fig 3A, B and S2I, J)? Since SGs are composed of RNPs, it seems obvious that their larger size implies the presence of more proteins and RNAs. Was there any other alternative? (ii) In lines 230-236, the description does not fit with Fig 5B and S3B: PBs were not counted prior to stress; after stress, Fig 5B shows smaller PBs following siDDX6 and si4E-T but not following siDCP1A, while Fig S3B shows twice more PBs after siDDX6 and si4E-T but not after siDCP1A; Fig 5B shows major changes in PB size after siLSM14A, though with high heterogeneity between cells. Moreover, since a similar study after silencing DDX6, 4E-T, DCP1A, PAT1B, LSM14A, EDC3 and arsenite treatment is already published (Ayache et al., 2015), potential differences should be discussed. (iii) In all violin plots, it would be useful to optimize the scale of the y-axis for better visualization. Moreover, indicating median values would be more relevant than mean values, which are easily biased by few cells with extreme patterns. (iv) How the authors interpret that siLSM14A reduces DCP1 as much as siDCP1A (Fig S3A) without affecting PBs like siDCP1A (Fig 5B)? (v) To which extent the SG enlargement is simply related to lower translation upon silencing of these various proteins? This is probably the case for siDDX6, in view of the literature and Fig S2H, and it would completely change the interpretation of the results. (vi) In Fig 5E, the effect of siLSM14A and siNOT1 on SG size in DDX6 KO cells is statistically significant but quantitatively minor, so that it seems difficult to draw any robust conclusion.

5/ The paragraph on PB-SG docking is particularly difficult to follow. (i) "All knockdowns led to an increase in the number of PBs and docking" (lines 260-264)... All knockdowns of which proteins? Increase compared to what? If the sentence applies to si4E-T and siDCP1A, how is it possible to draw general conclusions about the importance of homotypic vs heterotypic interactions from just these two cases? (ii) The discussion about the docking of "EDC4 spots" (lines 265-270) is not more convincing. Moreover, EDC4 immunostaining being finely granular in the cytosol, how is it possible to count accurately EDC4 spots and calculate docking (Fig S4B)? More generally, why not simply conclude that in the absence of PBs, PB components relocalize to SGs, as previously proposed? (iii) It has been previously reported that changes in PB-SG docking are linked to changes in SG size (Aulas et al., JCB 2015, PMID: 25847539). Can it explain some of the observations here?

6/ Concerning the rescue by DDX6 mutants, the description in lines 317-319 does not fit with Fig 6E and S5C, which only report PB count after stress. In lines 325-326, "These results suggest that DDX6 binding to EDC3, PAT1B, 4E-T, LSM14A are not required for PB rescue in DDX6 KO U-2 OS cells but rather for the formation of large canonical PBs" is unclear. Then, similarities and differences with previous publications should be discussed: it was already reported that the DDX6 E247A does not rescue PB assembly (Minshall et al., MBoC 2009, PMID: 19297524); only partial rescue was observed with DDX6 Mut1 in unstressed cells (Ayache et al., 2015); mutation in 4E-T demonstrated that 4E-T binding to DDX6 is important for PB assembly (Kamenska et al., NAR 2016, PMID: 27342281). Does it mean the results depend on the cell line?

Minor comments :

7/ What means a negative value in the puromycylation assays (Fig S2H)

8/ in line 198, Fig S3J instead of S2J. In line 319, Fig S5C instead of S5E?

9/ For DDX6 Mut1-4, indicate the mutated residues in the text. ATPase dead mutant cannot be E245A.

10/ In FigS4B, either the y-axis cannot go over 100 or it is not a percentage.

Reviewer #3 (Comments to the Authors (Required)):

The authors study the role of DDX6 and several other PB or SG proteins in RNP granule formation and docking. Importantly, the authors generate cell lines to systematically abrogate SG, PB, or both to systematically study the accumulation of proteins and RNA in the resulting granules. In the absence of DDX6, other PB components form puncture around, and some within, SG. The authors further study the effect of knockdown of other PB proteins - both in WT and in DDX6 KO cells - and study the effect on PB and SG formation. They identify some proteins that seem to work in complex with DDX6 (in particular the DDX6 - 4E-T - DCP1 interactions) and important to limit SG-PB formation. Other proteins have an additional effect to DDX6; overall changes in PB and SG formation and docking are not well correlated. Interestingly, for all knockdowns of PB proteins, they observe less PB but more docking between SG and PB. Independently of DDX6, also knockdown of CNOT1 or PAT1B reduces BP-SG docking both in WT and DDX6 KO cells.

In the last figure, the authors supplement the DDX6 KO cell with various DDX6 mutants that abolish RNA binding, ATPase activity or interaction with partner proteins. RNA binding and ATPase deficient mutants cannot restore PB formation, but 2 mutants deficient for EDC3 / 4E-T binding can. Interestingly, only one of the mutants rescuing PB formation can also rescue SG formation.

The experiments are in general cleanly described and labeled. Overall, the manuscript remains rather descriptive with limited mechanistic insight.

Specific comments:

- The authors describe (Figure 2) that DDX6 KO results in "limited growth and fusion" of SG. It would be interesting to see whether these structures have reduced turnover in FRAP.

- In Figure 2E and 2F, the PABPC1 cytoplasmic background seems to be higher in WT than in DDX6 KO cells. Could the

authors please quantify and comment? If true, this This could be interesting with respect to impaired turnover of SG.

- It would be helpful if the authors could quantify size (individual and total) and number of SG in Figure 2E, and test what other proteins apart from PABPC1 accumulate in triple-KO "SG" in Figure 2G (e.g. CNOT1 / PAT1B)
- In figure 3, the authors test accumulation of candidate RNAs in DDX6 KO cells. It would be interesting to perform this experiment also with the mutants described in Figure 6.
- The images in Figure 4, especially 4A, look very interesting, but the quality and resolution make it difficult to properly judge the them. A 3D version of tiger resolution could for example be very helpful for interpretation. Also combination with FISH (see Figure 3) would be informative to evaluate RNA localisation. The images of 4E-T are somewhat different between Figures 4A (looks more like a connected area on the surface of SG) and B (looks more 'central' and surrounded by EDC4), especially since only 1 cell is shown per experiment. It would be great if the authors could take a look at these figures again, and possibly add a three color staining with PABPC1 in 4B to help the reader orient themselves.
- not all PB knockdowns seem to be efficient. It would be helpful to quantify the knockdown efficiency and indicate it (in %) both in the supplementary figure western blots and in the main figure (IF) / text. Figure 5G/H/I: should this not be 'partition' coefficient?
- in Figure 6, an easy-to-grasp table with brief descriptions of the mutants would tremendously help the reader to understand the data.

Dear Roy,

Thank you for submitting your manuscript entitled "DDX6 modulates P-body and stress granule assembly, composition and docking" to the Journal of Cell Biology. Thanks also for your patience as it took longer than typical to provide you with a decision on your study. The manuscript has been evaluated by expert reviewers, whose reports are appended below. Unfortunately, after an assessment of the reviewer feedback, our editorial decision is against publication in JCB.

We are concerned that the work is mainly descriptive and lacks both mechanism and novelty, though the issues the experiments expose are topical and important. All three reviewers require improvements to the experiments themselves, which could support the characterization of the role of DDX6 in SG and PB dynamics. The quality of the imaging, including the extent to which molecular components can define SGs and PBs, was criticized. Most concerning are reviewer 1 and 2's comments that several of the introductory figures may reflect previously published experiments from several labs, including the Parker lab. They indicate that the literature has not been completely cited, giving credit to labs that have previously published experiments very closely aligned with these experiments. We hope that this situation will be remedied if you choose to resubmit (including a resolution of the overlaps with the work from Majerciak et al. now published as doi: 10.1093/nar/gkad585). Reviewer 2 suggests the possibility that some of the figures overlapping with published studies could be referenced and discussed in the introduction, allowing you to construct a manuscript that focuses on the more novel experiments that support a concise conclusion.

In this comment, the editor summarizes the reviews to make several points, which we address sequentially.

First, the editor notes that the manuscript lacks mechanism and novelty. We disagree since this manuscript provides new information on a role for DDX6 in limiting stress granules, describes a basis for protein accumulation in stress granules upon DDX6 loss, and provides insights into the regulation of P-bodies docking with stress granules. Given these contributions, we believe the manuscript has presented sufficient new information for publication.

Second, the editor suggests we make some technical improvements to the experiments. We have done so as described below.

Third, the editor suggests some of the initial figures are redundant with earlier work. While there is some overlap, we believe these are useful to a) reproduce other results, and b) define specific components of P-bodies and stress granules in U2OS cells, which has not been done with as much care (many studies looked at a few proteins in different cell lines), c) highlight cell type specific differences, d) addresses reviewer comments regarding lack of sufficient protein markers, and e) link to some of new key findings upon DDX6 KO. If needed, we would be willing to move this Figure to the supplemental but prefer not to do so since that will make the manuscript more onerous to read.

Fourth, the editor requests we cite additional prior work, which we have done.

Finally, the editor requests we resolve overlap with a related paper (Majerciak et al, 2023, published in NAR while our manuscript was under review.), which also comes to the conclusion that DDX6 functions to limit P-body and stress granule docking. However, we believe this should

not prevent the publication of our work for multiple reasons. First, the effect of DDX6 influencing P-body and stress granule docking is only a part of our manuscript. Moreover, we show this effect is independent of P-bodies per se, which is not addressed in the Majerckiak manuscript. Second, we come to different conclusions in other areas. For example, we suggest DDX6 limits stress granule formation, while Majerckiak et al., conclude that DDX6 promotes stress granule formation. Third, we do not believe we should be "scooped" on this subpoint of our work by a manuscript on Bioarchive. Now that their manuscript is published in a final form we have discussed and cited this work wherever appropriate.

Taken together, these revisions have improved our manuscript, and we trust it is acceptable for publication.

Therefore, while your manuscript is intriguing, I feel that the points raised by the reviewers are more substantial than can be addressed in a typical revision period. If you wish to expedite publication of the current data, it may be best to pursue publication at another journal. Our journal office will transfer your reviews upon request.

Given interest in the topic, I would be open to resubmission to JCB of a significantly revised and extended manuscript that fully addresses the reviewers' concerns including the presentation of new experimental work addressing all of their comments as well as scholarly treatment of the work in the context of previous published studies. If you would like to resubmit this work to JCB, you must contact the journal office to discuss an appeal of this decision. To place an appeal please prepare a detailed response to the reviewers along with a revision plan outlining how you will address all experimental issues. Please note that we may discuss an appeal with the original reviewers. Our goal with the appeal process is to ensure that planned revisions seem appropriate to potentially address all concerns before authors spend time and resources on revisions. Please note that priority and novelty would be reassessed at resubmission, and that a revised study would but subject to further peer-review by the original reviewers or suitable replacements.

We addressed all reviewer comments and significantly revised and extended our manuscript in a typical revision period. We believe the manuscript is now acceptable for publication.

Regardless of how you choose to proceed, we hope that the comments below will prove constructive as your work progresses. We would be happy to discuss the reviewer comments further once you've had a chance to consider the points raised in this letter. You can contact the journal office with any questions, cellbio@rockefeller.edu or call (212) 327-8588.

Thank you for thinking of JCB as an appropriate place to publish your work.

Sincerely,

Karla Neugebauer, PhD

Monitoring Editor

Andrea L. Marat, PhD
Senior Scientific Editor

Journal of Cell Biology

Reviewer #1 (Comments to the Authors (Required)):

Ripin et al. investigated the contribution of DDX6 to SG-PB docking, SG composition, and SG and PB formation. This is an important contribution to understanding the interactions between and dynamics of SGs and PBs, stress-responsive membraneless organelles. I am enthusiastic about this work and think it should ultimately be published, however additional work and revision are needed to provide sufficient evidence for the conclusions drawn and to appropriately set this work in the context of the current literature in the field.

We thank the reviewer for the enthusiasm about our work. We revised the manuscript and incorporated all the suggestions, which significantly improved the manuscript.

Major concerns:

1) Nomenclature: it would be helpful to understand the criteria for designating a condensate as a SG or a PB:

We have added specific criteria to designate an RNP granule as either a P-body or a stress granule in the introduction.

2) Since some of the phenotypes contain combinations of proteins that are typically restricted to SGs or PBs (i.e. the SGs in DDX6 KO cells treated with arsenite), it does not seem appropriate to call these SGs or PBs, as they seem to represent some kind of fusion of the two. Clarify the discussion of these

Based on our data, we do believe these are distinct granules where some PB proteins form smaller assemblies that still interact with SG and not just a fusion, which is supported by our 3D imaging added to this revision. We have clarified this issue and how we refer to the different types of RNP granules in the text.

3) Immunofluorescence images lack sufficient protein markers (or use inappropriate markers) to unambiguously assign them as SGs or PBs (more markers will help address above point). In particular, the following figures should be addressed: Fig 2 A, E, F, G; Fig 4 B, C; Fig 6 C, D; Fig S2 F. Please see details in "additional comments" section below.

We tested PABPC1, eIF4A, G3BP1, HuR, TIA1, YB1, eIF4G1, eIF4E, UBAP2L, DHX36, Staufen, oligodT-RNA FISH, all which are considered in the literature as appropriate SG markers and DCP1, EDC4, EDC3, 4ET, LSM4, LSM14, CNOT1 as PB markers (some are shared), to unambiguously assign them as either unique (or shared) SG or PB marker in U2-OS cells, many for them are shown in figure 1 (not all due to redundancy). All showed the same results when comparing WT, DDX6 KO and triple KO cells (addressing Fig. 2 A, G; Fig. 4 A,B,C, Fig S2 F). We didn't include all markers in the manuscript, but added more examples to adequately address the reviewer comments. Similarly, using many PB markers, we show relocalization and/or clustering around SGs in the DDX6 KO (addressing Fig. 4 A,B,C).

From our data, we choose PABPC1 and EDC4 as SG and PB marker as these have been commonly used in the field before, are observed unambiguously in SGs or PBs, respectively, as shown in our images, and are not affected by DDX6KO, G3BP1/2 KO or siRNA KDs that we performed.

We now added more IF examples, quantification of another marker and emphasized more strongly in the text that we tested many markers and why we chose to continue with PABPC1 and EDC4 as the main markers. We also added another PB marker in this experiment (Fig 6 C, D), which addresses the reviewer's comment below.

4) Work needs to be placed in context of additional relevant publications:

The reviewer asks us to place our work in the context of two earlier papers. We have done so. Specifically, we:

A) Mention the work from Stoecklin and Kedersha suggesting that overexpressed TTP1 can play a role in docking P-bodies and stress granules (in addition to the interactions we describe). However, whether that is relevant to normal conditions when TTP1 is not over-expressed has never been examined.

B) Discuss how the multiphase model of RNP granules applies to P-bodies and stress granules based on our results and those in Sanders et al., 2020.

5) Several conclusions regarding how DDX6 contributes to SG composition, docking etc. depend on the assumption that the DDX6 mutants share the phenotypes of the analogous DHH1 mutants (ie impaired ATPase activity or RNA binding). The effect of these mutations on DDX6 activity should be validated in human cells, or at the least the DDX6 and DHH1 sequence alignment should be included to show how similar the sequences are and increase the confidence in the conclusions drawn.

We thank the reviewer for the suggestion. We made a the DDX6 and DHH1 sequence alignment (now added to the supplemental figures), which demonstrates strong sequence and structural conservation, which we use to infer conserved function.

Additional comments:

Results:

6) loss or restoration of constitutive PBs: U2OS are not the best model for this, as few U2OS cells have any constitutive PBs. Would be better to perform in HeLa or Cos7 cells.

In principle, it would be interesting to repeat many of these experiments in additional cell lines. However, we believe the results we have are sufficient to make our conclusions and the work required to generate the necessary cell lines is not possible within a reasonable time frame.

7) Fig 2A,E,F: SG quantification based on only one SG marker (PABPC) - should have at least one additional SG marker (G3BP1, eIF3b, or eIF4A).

We have tested multiple SG markers (see comment above) comparing SG numbers in WT and DDX6 KO cells, added TIA1, UBAP2L and G3BP IF images (Fig. 2A), and added G3BP1 quantification as an additional example (Fig. S2F).

8) Are these canonical SGs? Based on figure 4, they are not since they contain PB proteins, thus it seems inappropriate to call them SGs.

The reviewer is raising a semantic issue about whether stress granules that contain P-body proteins under some conditions are still stress granules. However, we believe these are still stress granules since they only form during stress and are composed of many tested stress granule proteins. PB proteins that “fully” relocate into stress granules under some mutant conditions, show already enrichment in stress granules even in WT cells (e.g. CNOT1, LSM4, LSM14 and GW182 (Majerciak et al , 2023 NAR), while PB proteins only enriched in PBs (e.g. EDC4, DCP1) form rather smaller assemblies that dock to stress granules or form subassemblies at surface/center (e.g. 4E-T). We have clarified this point in the text.

9) In the text, PBs are described as docking to SGs in DDX6 KO cells (lines 204-222, Fig 4A, B). To me, 'docking' suggests two entities making contact at their edges. In WT conditions I agree that the PBs are docking at SGs, as typically observed under arsenite. However, in the case of the DDX6 KO cells 'docking' does not seem to accurately describe the relationship between SGs and PBs, as many PB-like puncta are not at the periphery of the SG but appear to be inside of it. While some nuance to the type of PB-SG association is discussed the description should be modified for clarity. See related comment above about definitions as well.

We have clarified the types of interactions we observe and how we refer to them. We also added 3D images to better highlight that those spots are indeed docking to the SG surface.

10) Fig 2D - Since the "SGs" in DDX6 KO cells contain PB proteins (per Fig 4), it is conceivable that they are actually a mixture of SG and PB components - therefore, it would be informative to include a count of area covered by SGs and PBs (collectively) in WT cells for comparison. It may be that this will show no difference between WT and DDX6 KO.

We have performed the suggested analysis (Fig. S2 G). We obtained the same observation as before (Fig. S2 G vs S2F). This suggested analysis nicely rules out that the change in SG parameter is only due to a mixture of SG and PB components.

11) Lines 174-176/ Fig 2E,F : since granules in the DDX6 KO cells contain both PB and SG proteins, the difference in their formation and disassembly rates from WT U2OS SGs could be due to a PB-like composition. From your data (Fig 1B) and that of others, we know that PBs form earlier than SGs in response to stress. Therefore, given that the granules in DDX6 KO cells

contain PB proteins, it could be that their rapid formation and delayed disassembly is due to a PB-like composition.

Including a PB marker in these experiments would allow for comparison to PB assembly and disassembly rate in WT cells to make this distinction between DDX6 "limiting SG assembly" or resulting in PB-like granules that inherently form more quickly and disassemble more slowly.

We thank the reviewer for this suggestion. We have added a figure with a PB marker for earlier time points during the time course (Fig. S3 E,F, EDC4 signal overamplified for better visualization of potential smaller PB like assemblies). Compared to WT cells, where PBs form earlier, no PB like assemblies can be observed at earlier points, highlighting an essential role of DDX6 in PB formation. In DDX6KO cells, SGs seem to form first, then recruit smaller PB like assemblies. This is nicely seen in some cells with smaller SGs but less docking of smaller PB like assemblies (Fig, S3 E, bottom). This is excluding the function of a PB-like composition in earlier SG formation. Together with the siRNA KD experiments for other PB factors and DDX6 mutants, this data nicely supports the conclusion that the DDX6 protein is limiting SG assembly and not the PB or PB like clusters.

Moreover, we added the data for disassembly and while PBs still persist in WT cells, smaller PB like assemblies are only visible docked to SG. Cells without SG don't show any PB like assemblies, supporting our hypothesis.

12) Fig 2G/S2 F - need WT cells and additional SG and PB markers on the same sample to examine co-localization and possible formation of SGs (or SG-like granules) in the triple KO.

The corresponding WT cells for 2I (original 2G) are shown in Fig. 2A. We have separated the figures to highlight fewer but key points and for a better flow. We added this information to the figure legend. We now added the corresponding PB marker into fig. S2J to again focus on the flow/ the main points. We also added another SG markers as asked by reviewer 3 and to address another comment from below regarding unhealthy cells.

13) Line 317: 'rescue' of PBs is misleading since few cells have PBs in the absence of stress in U2OS cells. "nucleation" would be more accurate.

We rephrased the sentence.

14) Conclusions from the DDX6 mutant experiments largely depend on the assumption that the mutations in DDX6 have the same effect as in yeast DHH1. The mutations should be validated in order to draw these conclusions. E.g. - does mut 4 impair ATP hydrolysis activity?

As suggested by the reviewer, we have pointed out that the conservation of DDX6 and DHH1 (including the critical functional residues shown in Fig S5A) allows us to infer the effect of the mutations in mammalian cells. We have noted the limitations of our inferences.

15) Figure 4 D,C: granules referred to as SGs, but no SG marker shown.

We have rephrased this section.

16) Figure 6C, D: additional PB markers need to be included to confirm presence or absence of PBs in cells expressing the different DDX6 mutants. It is possible that some DDX6 mutants may

not be recruited to PBs, but PBs may still form. Or, DDX6 aggregations may form, but other PB proteins are not recruited, and thus PBs have not formed.

We agree and we have added data (S5 E) showing another PB component gives the same result.

17) contrast should be increased on many images as they are difficult to see (e.g. Fig S2 D, Fig S4 A, Fig S5 C).

As suggested, we increased the contrast.

18) Fig S2 E - bottom right panel - no cells?

We fixed the figure, that had the DAPI channel missing.

19) Fig S2 F G1/G2/DDX6 KO arsenite trmt: is it possible these cells are unhealthy and the eIF4a staining reflects how any protein might look? Need non-SG control protein or additional SG protein marker.

We now show additional markers such as PABP in Fig. 2I, and YB1 and UBAP2L in Fig. S2I. The cytoplasm just looks granular due to the formation of smaller SG like foci. Moreover, although weak, the puromycin stain in Fig S2 L (non-SG control) shows uniform distribution.

The cells look healthy to us and don't show any significant changes in terms of nucleus or cell shape that would indicate unhealthy cells.

20) Fig 2G: need WT control (will show arsenite and PABPC antibody behaving as expected).

Comment addressed above.

Methods:

21) Table S1 not included, unable to fully assess methods.

The table S1 was added an excel file, which failed to properly export into the PDF. We have now added this table as a normal table in word format.

22) 500 μ M arsenite for 1 hr is a very strong dose. Do these phenotypes occur at lower arsenite doses or in response to other stresses? Or does the system have to be pushed to its max to get these results?

500 μ M is very commonly used for stress granule analysis (by our lab and many others) and is not the maximum used in the literature. Moreover, in our analyses, we also used 100uM for the recovery experiment (and saw the same type of results). In addition, we have examined stress granule formation in the presence of hippuristanol or sorbitol and get the same results comparing WT and DDX6KO (Fig. 2. F, S2H).

23) It is surprising that 1:100 dilution was needed to see DDX6 in SGs. DDX6 localization to SGs and PBs has already been documented.

We believe the 1:100 dilution could have led to an unspecific staining. We tested a second antibody- from proteintech at 1:1000 and 1:100 and could also not see staining in SG. We therefore removed that panel from the supplementary material and replaced it with the new observation.

There are divergent results in the literature as well. For example, Mollet et al., 2008 showed DDX6 in SG in HeLa cells while Kedersha and Anderson, 2007 showed DDX6 is not detected in SG in U2OS cells. These divergent results are part of the reason we think our basic characterization experiments in the beginning of the manuscript are useful to the field. We provided a hypothesis for these cell type specific effects in the discussion.

24) Miscellaneous minor points:

We thank the reviewer for catching these minor issues, all of which have been fixed.

- Ugay affiliation missing **fixed**
- Line 157 "cells" \diamond "cell" **fixed**
- Fig 6B and Fig S5A: DAED box \diamond DEAD box **fixed**
- Several places throughout MS where SG or PB should be plural. **fixed**
- Labeling of extended and supplemental figures is confusing and makes it difficult to find the correct figures. For example, in the text Fig. S3 J and H are referenced. However, Fig. S3 does not have "J" or "H" and is not the correct figure. Rather, "Extended Fig. 3" should be referenced, which is contained in "Fig. S2". Even with the referencing corrected, this remains confusing, as there is no clear distinction within Fig. S2 between Extended Fig. 2 and 3. Please revise. **fixed**
- Fig 5 is referenced in the text prior to figure 3. Please revise so figures are references in the order they are presented. **fixed**
- Line 153: Fig 2A referenced for triple KO but that is not part of the figure. **Fixed**.
- Are Fig 6B and Fig S5A the same? Please correct/clarify.
Yes, it is the same to illustrate the rotation and removal of other components. We indicated this in the figure/legend.
- Fig S1 F: labels missing – which protein is each graph for?
We added the protein names.

Reviewer #2 (Comments to the Authors (Required)):

This study reports that the RNA helicase DDX6, a known P-body (PB) component, can also limit the formation of stress granules (SG) in an ATPase-dependent manner, and that several PB proteins can influence the composition of SGs and their docking with PBs. The experiments show how DDX6, wt or mutant, and a few other proteins affect PB and SG number and size, as well as PB-SG docking. They involve immunostaining of classical PB and SG markers and a few single molecule RNA FISH, in a single cell line (U-2 OS) upon a single stress (arsenite).

At this stage, the manuscript presents well-conducted experiments but of limited novelty. First, that silencing of some PB proteins affects the composition of SGs has been already reported in the literature. Second, the impact of these proteins on SG size was extensively described by Majerciak et al. in a preprint deposited on BioRxiv (2021). Furthermore, it seems difficult to build a strong model, based only on variations of SG size and PB-SG docking, particularly when the effects are significant but quantitatively minor.

We disagree with the reviewer that the manuscript lacks mechanism and novelty. His/Her first point is that silencing of some PB proteins affects the composition of SGs. While this is true, we provide additional information that clarifies this process with three key points. First, proteins

that already show some partitioning in SGs in WT cells, show enhanced partitioning into SGs when PBs are limiting. Second, by using newer microscopes with enhanced resolution, we show that proteins unique to PBs do not partition into SG, but instead form small PBs that dock to the SG surface. Third, we show that this is influenced by how DDX6 binds RNA and hydrolyzes ATP. Taken together, we have made important new contributions to our understanding of how proteins partition between PBs and SG, and how each granule can affect the other.

The reviewer's second point is that the impact of DDX6 on SGs has already been shown by another group. We do not believe this prior manuscript should limit our work's publication since we come to different conclusions with Majerciak concluding that DDX6 promotes SG, while our work suggests DDX6 limits stress granule formation. Finally, we do not think a manuscript should be rejected because of a Bioarchive manuscript. In our revisions, we have clarified how our work relates to the Majerciak work.

Given these contributions, we believe our manuscript has presented sufficient new information for publication.

Major concerns:

1/ Since the experiments described from lines 97 to 148 essentially replicate data published by the Parker (Buchan et al., 2008, Jain et al., 2016) and other groups (Mollet et al., 2008, Kedersha and Anderson, 2007; Hubstenberger et al., 2017; Kedersha et al., 2005), these results could be all described in the introduction.

We agree that some of these experiments were performed before, and we referenced those findings in our first version. However, we disagree that lines 97 to 148 are essentially replicate data. This is one of the few experiments that have directly compared all these proteins under the same conditions in the same cell line. Moreover, unlike earlier work, we provide careful partition coefficients that illuminate protein distribution between SG and PBs and are important in the interpretation of the phenotypes in the DDX6 KO cell lines.

Taking the reviewer's comment to heart, we have shortened this section. Moreover, if desired, we could move this figure to the supplement, but prefer not to facilitate reading of the manuscript.

Reference should be added for the well-known observation that PABPC, G3BP1 and polyA RNA are not enriched in PBs (line 122).

We have added the requested references.

And for the fact that DDX6 is more abundant than other PB proteins (line 144) (a Table in Ayache et al., 2015, includes the analysis presented in Fig S2A).

We agree and had referenced Ayache et al 2015 in this section before. We believe that the supplemental tables shown in Ayache et al. (which give the relative abundance of proteins as a fraction of DDX6) are not as useful as absolute numbers of molecules, which we show in our figure. Moreover, we are not making the point that DDX6 is more abundant than other PB proteins, as we were referencing other papers, but rather that this observation suggests more unknown functions for DDX6 besides making PBs. We rephrased this section to make this clearer.

2/ What is the positioning of the authors with respect to the Majerciak et al preprint (BioRxiv 2021). This preprint is entitled "RNA helicase DDX6 in P-bodies is essential for the assembly of stress granules". It addresses the exact same question, with similar approaches. It already describes that the number of arsenite-induced SGs increases after DDX6 silencing, in an ATPase-dependent manner (using the same E247A mutant). Yet, surprisingly, this preprint is cited only once in the result section, for a minor reason (line 386), and not at all in the discussion.

The reviewer unfortunately missed that we were citing the paper early in the introduction (original line 58) and said that DDX6 "was recently suggested to affect SG formation (Majerciak et al., 2021)". It should be noted that the original preprint came to the opposite conclusion as our work, contained less experimental approaches, less data and almost no quantifications. Now that this work is published in an updated manner, we have added more discussion of the work in our manuscript.

3/ Throughout the manuscript there are ambiguities between organelles, organelle-like foci/assemblies and organelle components, which weaken the conclusions.

(i) In G3BP1/2 KO cells, which are devoid of SGs, DDX6 KO leads to the reappearance of "SG-like foci", while the conclusion of the paragraph is about DDX6 limiting the formation of "SGs" (lines 184-186). In view of their completely different morphology, what is the link between "SG-like foci" and SGs beyond the fact they share PABPC1 and eIF4A (Fig 2G and S2F)?

These are smaller stress granules since they only form during stress and contain several stress granule proteins. Moreover, similar experiments on the helicase eIF4A or ADAR1 lead to the same observations where the link between "SG-like foci" and SGs has been extensively characterized (Tauber et al 2020, Corbet et al 2021). We have now added more SG markers and made this inference explicit in the text.

(ii) Following the silencing of DDX6, which abolishes PBs, some PB proteins accumulate at the periphery of SGs, in a pattern described as "PB-like assemblies" (lines 205-210), while others localize in both the periphery and the central region of SGs (lines 218-219). Probably by mistake, 4E-T is included in the two classes.

We extended our explanation in the text to describe the three types of changes. Proteins that accumulate at the periphery of SGs, localize in both the periphery and the central region of SGs and which fully enter SGs. 4E-T seems to localize in both the periphery and the central region of SGs. We performed super resolution microscopy and added new data that supports that conclusion.

(iii) More important, when docking is analyzed, the conclusion is that DDX6 reduces "PB"-SG docking (lines 283-284), despite the absence of PBs in these conditions.

We rephrased this section.

(iv) It was already reported that EDC4 and DCP1 relocalize in SGs when PBs are abolished using siDDX6 (Mollet et al., 2009; Serman et al., NAR 2007, PMID: 17604308), except that these authors did not observe a stronger accumulation at the SG periphery. Does it depend on the cell line? Or on the efficiency or timing of the silencing?

We believe the difference is the degree of resolution in the imaging and the interpretation of the images. Both Serman et al., 2007 and Mollet et al 2008 used a Leica DMR microscope with a 63X1.32 oil immersion objective, while we use a 100x oil immersion objective with 1X or 2.7x magnification on the Nikon Spinning Disk Super Resolution by Optical Pixel Reassignment (SoRa) microscope. Despite lower resolution on the DMR microscope, in both Serman et al., 2007 and Mollet et al., 2008, some images clearly show distinct DCP1 foci at the SG surface. One example is shown below (Serman et al, Fig. 6F): siRNA KD of DDX6 shows how DCP1 foci form smaller punctate assemblies at SGs with some strong signal/enrichment at the SG periphery.

We clarified this better in the manuscript and highlight/cite the Serman et al., 2007 and Mollet et al 2008 observations in our manuscript.

4/ The study of SG size following silencing of various factors raises several questions.

(i) What do we learn from the smFISH experiments (Fig 3A, B and S2I, J)? Since SGs are composed of RNPs, it seems obvious that their larger size implies the presence of more proteins and RNAs. Was there any other alternative?

We believe showing increased mRNAs is useful since a) when you see more protein in a SG, it is an inference that RNA is also increased, and it is useful to directly demonstrate that increase, and b) in prior work with inhibition of eIF4A mRNA specific effects of accumulation in stress granules was observed, which we were also testing in this experiment.

(ii) In lines 230-236, the description does not fit with Fig 5B and S3B: PBs were not counted prior to stress; after stress, Fig 5B shows smaller PBs following siDDX6 and si4E-T but not following siDCP1A, while Fig S3B shows twice more PBs after siDDX6 and si4E-T but not after siDCP1A; Fig 5B shows major changes in PB size after siLSM14A, though with high heterogeneity between cells. Moreover, since a similar study after silencing DDX6, 4E-T, DCP1A, PAT1B, LSM14A, EDC3 and arsenite treatment is already published (Ayache et al., 2015), potential differences should be discussed.

1) We removed the statement for the non-stressed conditions. 2) We modified the text for better clarity. 3) We discuss the differences and similarities compared to Ayache et al., 2015.

(iii) In all violin plots, it would be useful to optimize the scale of the y-axis for better visualization. Moreover, indicating median values would be more relevant than mean values, which are easily biased by few cells with extreme patterns.

Although we agree with the reviewer in principle, we have not increased the size of the violin plots axis since the violin plots have multiple conditions in one figure and a smaller size is necessary to keep all the plots consistent throughout the manuscript.

At the beginning of our study, we tested both, median and mean and obtained the same conclusions. We therefore decided to go with the mean, as this is still more commonly used compared to the median.

(iv) How the authors interpret that siLSM14A reduces DCP1 as much as siDCP1A (Fig S3A) without affecting PBs like siDCP1A (Fig 5B)?

In retrospect we picked an unrepresentative western to illustrate the siLSM14a effect, which we have corrected. Rerunning the same samples and other biological replicates did not show an effect of siLSM14 on DCP levels. We corrected the western blot.

(v) To which extent the SG enlargement is simply related to lower translation upon silencing of these various proteins? This is probably the case for siDDX6, in view of the literature and Fig S2H, and it would completely change the interpretation of the results.

We tested this possibility by examining the translation rates in WT and DDX6KO during stress and observed that translation silencing is unchanged between WT and DDX6KO (Fig. S2 L and quantification in M, right side).

(vi) In Fig 5E, the effect of siLSM14A and siNOT1 on SG size in DDX6 KO cells is statistically significant but quantitatively minor, so that it seems difficult to draw any robust conclusion.

We agree and have rephrased our conclusions appropriately. As usual, siRNA KD experiments don't have a 100% efficiency (see new western blot quantifications for our efficiencies in Fig. S4 A), thus, all the statistics are done on a mixed population. This can often lead to the quantitatively minor effects.

5/ The paragraph on PB-SG docking is particularly difficult to follow.

(i) "All knockdowns led to an increase in the number of PBs and docking" (lines 260-264)... All knockdowns of which proteins? Increase compared to what? If the sentence applies to si4E-T and siDCP1A, how is it possible to draw general conclusions about the importance of homotypic vs heterotypic interactions from just these two cases?

We have edited this paragraph to make it clearer.

(ii) The discussion about the docking of "EDC4 spots" (lines 265-270) is not more convincing. Moreover, EDC4 immunostaining being finely granular in the cytosol, how is it possible to count accurately EDC4 spots and calculate docking (Fig S4B)?

We do not fully understand this comment. In our work, we clearly see a punctate EDC4 (and of other proteins) signal in DDX6 KO cells. Given this, we were able to quantify the images using standard approaches as regularly done for smFISH RNA spots.

More generally, why not simply conclude that in the absence of PBs, PB components relocate to SGs, as previously proposed?

While this has been previously proposed, our images are more consistent with two types of protein redistributions. First, proteins that are unique to PBs form smaller PB-like assemblies that then dock to the surface of stress granules. Second, PB components that can also be seen in SG (such as 4E-T, LSM14a, etc) now show some relocalization into SG, presumably due to their pre-existing interactions with SG proteins or RNAs. We have clarified these points in the text.

(iii) It has been previously reported that changes in PB-SG docking are linked to changes in SG size (Aulas et al., JCB 2015, PMID: 25847539). Can it explain some of the observations here?

Aulas et al 2015 showed reduced SG size upon siRNA KDs that correlate with PB-SG docking. However, an important other explanation was not addressed. Which is that the proteins themselves are mediating key interactions and KD of these RBPs reduces homotypic interactions (SGs gets smaller) and heterotypic (less docking). No experiments were done to distinguish loss of protein function vs loss of PBs. If solely the granule size would influence docking, this should have been also true for smaller SGs in the negative control as well. This rather suggests that the proteins are regulating docking instead of the SG size.

We also saw similar correlations and first wondered if the granule size (SGs or PBs) influences docking. However, SG size alone, as shown in our Fig.5 and S4 is not mediating docking in our experiments. We therefore interpret our data that rather specific protein-protein or protein-RNA interactions mediate docking.

6/ Concerning the rescue by DDX6 mutants, the description in lines 317-319 does not fit with Fig 6E and S5C, which only report PB count after stress.

We are not clear about the suggestion. We indicated "Fig. 6 C, S5 C", in line 317-319 which show PBs upon rescue in stressed (6 C) and unstressed (S5 C (now D)) cells. We nevertheless rephrased that section and hope this issue is resolved.

In lines 325-326, "These results suggest that DDX6 binding to EDC3, PAT1B, 4E-T, LSM14A are not required for PB rescue in DDX6 KO U-2 OS cells but rather for the formation of large canonical PBs" is unclear.

We have edited this sentence to clarify our point.

Then, similarities and differences with previous publications should be discussed: it was already reported that the DDX6 E247A does not rescue PB assembly (Minshall et al., MBoC 2009, PMID: 19297524); only partial rescue was observed with DDX6 Mut1 in unstressed cells (Ayache et al., 2015); mutation in 4E-T demonstrated that 4E-T binding to DDX6 is important for PB assembly (Kamenska et al., NAR 2016, PMID: 27342281). Does it mean the results depend on the cell line?

We think some of these results depend on different cell lines, knockdown efficiencies but also on image resolution (see explanation above). We have added a more extensive discussion of the literature including the differences between different experiments.

Minor comments :

7/ What means a negative value in the puromycylation assays (Fig S2H).

This is an error in the graphing, which we have corrected.

8/ in line 198, Fig S3J instead of S2J. In line 319, Fig S5C instead of S5E?

We corrected this error.

9/ For DDX6 Mut1-4, indicate the mutated residues in the text.

We have done so.

ATPase dead mutant cannot be E245A.

We fixed this typo.

10/ In FigS4B, either the y-axis cannot go over 100 or it is not a percentage.

We have corrected this error in the graphing.

Reviewer #3 (Comments to the Authors (Required)):

The authors study the role of DDX6 and several other PB or SG proteins in RNP granule formation and docking. Importantly, the authors generate cell lines to systematically abrogate SG, PB, or both to systematically study the accumulation of proteins and RNA in the resulting granules. In the absence of DDX6, other PB components form puncture around, and some within, SG. The authors further study the effect of knockdown of other PB proteins - both in WT and in DDX6 KO cells - and study the effect on PB and SG formation. They identify some proteins that seem to work in complex with DDX6 (in particular the DDX6 - 4E-T - DCP1 interactions) and important to limit SG-PB formation. Other proteins have an additional effect to DDX6; overall changes in PB and SG formation and docking are not well correlated. Interestingly, for all knockdowns of PB proteins, they observe less PB but more docking between SG and PB. Independently of DDX6, also knockdown of CNOT1 or PAT1B reduces BP-SG docking both in WT and DDX6 KO cells. In the last figure, the authors supplement the DDX6 KO cell with various DDX6 mutants that abolish RNA binding, ATPase activity or interaction with partner proteins. RNA binding and ATPase deficient mutants cannot restore PB formation, but 2 mutants deficient for Edc3 / 4E-T binding can. Interestingly, only one of the mutants rescuing PB formation can also rescue SG formation. The experiments are in general cleanly described and labeled. Overall, the manuscript remains rather descriptive with limited mechanistic insight.

We appreciate that the reviewer feels the manuscript is generally well done. We disagree with this reviewer that the manuscript lacks mechanism and novelty. We disagree since this manuscript provides new information on a role for DDX6 in limiting stress granules, defines the

interplay between P-body and stress granule protein partitioning, and identifies P-body proteins that promote docking with stress granules. Given these contributions, addressing the following comments and incorporating various suggestions, we believe the manuscript has presented sufficient new information for publication.

Specific comments:

1) The authors describe (Figure 2) that DDX6 KO results in "limited growth and fusion" of SG. It would be interesting to see whether these structures have reduced turnover in FRAP.

As suggested by the reviewer, we added this experiment to the manuscript (Fig. S2 K) and observed no significant difference in G3BP dynamics in SG comparing WT and DDX6 KO cell lines.

2) In Figure 2E and 2F, the PABPC1 cytoplasmic background seems to be higher in WT than in DDX6 KO cells. Could the authors please quantify and comment? If true, this This could be interesting with respect to impaired turnover of SG.

The reviewer raises an interesting point. We have quantified the PABPC1 and oligo(dT) FISH signal and observe they are both lower in DDX6KO cells. We believe this is due to a change in the average poly(A) tail length, which at least in yeast *dhh1Δ* strains shows a shift to shorter A tails (Coller et al., 2001, RNA). We have chosen not to discuss this issue more since the impact on differential A-tail length on P-body and stress granule formation is an entire separate manuscript.

3) It would be helpful if the authors could quantify size (individual and total) and number of SG in Figure 2E,

We have done the quantification and see an increase in the number, area and total SG area over time. We have chosen not to add this data since detailed SG parameter quantifications (number and or volumes) during a time course of stress have been already done in the past (e.g. Wheeler et al 2016, Guillén-Boixet et al 2020) and our analysis is not providing any new information. The point of this experiment was to assess if there was any substantial alteration in the time course comparing WT and DDX6 KO cells.

and test what other proteins apart from PABPC1 accumulate in triple-KO "SG" in Figure 2G (e.g. CNOT1 / PAT1B).

We have added more examples into the manuscript. E.g. eIF4A into Fig 2I and YB1 and UBAP2L as examples into Fig S2I. Moreover, we added EDC4 as PB markers into Fig S2J.

4) In figure 3, the authors test accumulation of candidate RNAs in DDX6 KO cells. It would be interesting to perform this experiment also with the mutants described in Figure 6.

In Figure 3, we performed the smFISH to determine if there was any specificity in the mRNAs accumulating in stress granules in DDX6 KO cells. We observed all mRNAs tested showed an increase. Given this, we do not think additional smFISH experiments in the various DDX6 mutants will yield significant information.

5) The images in Figure 4, especially 4A, look very interesting, but the quality and resolution make it difficult to properly judge the them. A 3D version of tiger resolution could for example

be very helpful for interpretation. Also combination with FISH (see Figure 3) would be informative to evaluate RNA localisation.

As requested, we have added a 3D images for EDC4, DCP1 and 4ET using Imaris for better visualization (Fig. 4B, Fig.S3C).

For the three RNAs reported in this study, we extensively characterized RNA location in SGs before (Khong et al 2016, JCB, MollCell 2018). Thus, we are not sure what additional information we should provide.

The images of 4E-T are somewhat different between Figures 4A (looks more like a connected area on the surface of SG) and B (looks more 'central' and surrounded by EDC4), especially since only 1 cell is shown per experiment. It would be great if the authors could take a look at these figures again, and possibly add a three color staining with PABPC1 in 4B to help the reader orient themselves.

We have examined multiple images of these experiments. We noticed that 4E-T docks to the surface (Fig. 4A,B) but some spots also localize to the center, showing inhomogeneous distribution. Therefore 4E-T seems to behave differently compared to DCP1 and EDC4. To address this, we have used better magnification and added those images to Fig. 4F. Here, 4E-T shows subassemblies, indicating it can localize at the surface and the center rather than having homogeneous SG localization, explaining the observations in the original images. We added detailed explanations to the text.

- not all PB knockdowns seem to be efficient. It would be helpful to quantify the knockdown efficiency and indicate it (in %) both in the supplementary figure western blots and in the main figure (IF) / text.

As suggested, we have shown the efficiencies of the knockdowns in the western blot fig. S4A and mention the efficiencies in the text.

Figure 5G/H/I: should this not be 'partition' coefficient?

We have fixed the typo.

- in Figure 6, an easy-to-grasp table with brief descriptions of the mutants would tremendously help the reader to understand the data.

We hope Table 1 will address this suggestion.

February 1, 2024

Re: JCB manuscript #202306022R-A

Dr. Roy Parker
University of Colorado Boulder
Dept of Biochemistry
JSCBB
Boulder, CO 80309

Dear Dr. Parker,

Thank you for submitting your revised manuscript entitled "DDX6 modulates P-body and stress granule assembly, composition and docking". The manuscript has been seen by the original reviewers whose full comments are appended below. While the reviewers continue to be overall positive about the work in terms of its suitability for JCB, some important issues remain.

As you will see, while the reviewers appreciate that your study has been much improved a few important points remain. Regarding the level of advance, as your study was submitted to JCB before the Majerciak paper was published in a peer reviewed journal, our policy is that it does not detract from the novelty. However, while we appreciate that you have added in additional discussion of the Majerciak paper we agree that you should take the opportunity to discuss and synthesize prior work on this topic to present the reader with an overall current view. This seems especially important, since reviewer 1 mentions confusion about earlier literature regarding the idea that DDX6 is either more concentrated or exclusively localized to one or the other condensate (in particular, with reference to Kedersha and Anderson 2007). Perhaps you could discuss here that the 3D visualization in Fig 4 does not rule out presence of components on the inside of SGs. In Figure S2J you must include untreated cells as requested. Adding double staining with PABPC1 in S5E is not essential. Given your previous response to beginning your study with a recapitulation of what is mainly published data, we respect your choice of presentation. Finally, in your final revision please carefully respond to and address all other remaining reviewer concerns.

Our general policy is that papers are considered through only one revision cycle; however, given that the suggested changes are relatively minor we are open to one additional short round of revision.

Please submit the final revision within one month, along with a cover letter that includes a point by point response to the remaining reviewer comments.

Thank you for this interesting contribution to Journal of Cell Biology. You can contact me or the scientific editor listed below at the journal office with any questions at cellbio@rockefeller.edu.

Sincerely,

Karla Neugebauer, PhD
Monitoring Editor

Andrea L. Marat, PhD
Senior Scientific Editor

Journal of Cell Biology

Reviewer #1 (Comments to the Authors (Required)):

Overall, the manuscript has been much improved by the revisions. The manuscript is much clearer now in terms of defining what constitutes a SG or PB, and the rationale for the markers they use. The authors also included several experiments proposed by the reviewers which strengthen their claims. The manuscript will be strengthened by a little more discussion of how this work complements and/or contradicts the Majerciak paper, as well as addressing a few other minor comments.

- While the authors have added more discussion on Majerciak, one of the key points is not mentioned, which is that they propose that DDX6 is essential for the separation (creating the normal docking) of SGs and PBs. It would be appropriate to mention this in the discussion paragraph on docking.

- Fig S2J is a good addition, however it is so small it is hard to see what they authors are hoping to show with it. It would also help to include the same call line but untreated (without arsenite). I think this is important here because the assemblies are not as distinct as in WT cells, so comparing the images to untreated cells may help make the point stronger.
- Fig S5 E: including PABPC1 here would make it easier for the reader to compare to the corresponding figure in the main text (Fig 5). Please also increase the brightness of the image so EDC4 is clearer.
- Fig 4 - 3D visualization clearly shows that PBs are on the outside of SGs, but does it rule out their presence inside?
- Confusion over whether DDX6 is in PBs or not: Line 141-142: this is incorrect. Kedersha and Anderson 2007 show greater DDX6 (P54/RCK) recruitment to PBs than SGs, but not that it is only in PBs. Line 161-162 supports DDX6 in SGs.
- Line 262 - 263: should this be "enrichment in SGs and PBs in WT cells"?
- Line 275: components, not comments
- Should be Brigham and Women's Hospital

Reviewer #2 (Comments to the Authors (Required)):

The authors have now appropriately cited the previous literature, including the Majercki et al. recent publication. Compared to some of this work, I agree with the authors that the quality of imaging is increased by the use of newer microscopes. Yet, I believe the novelty of the conclusions is too incremental for the Journal of Cell Biology. I also maintain it is difficult to build a strong model based only on variations of SG size and PB-SG docking, particularly when these variations are quantitatively minor.

Some of my previous concerns were not fully addressed:

1/ Beginning of the results that essentially replicate published data. (i) I agree with the authors that they need to verify the presence of the various proteins they will study in SG and in PBs in U-2 OS cells, since differences between cell lines were reported in the past. However, for the reader who is eager to see new data, it would be more efficient to give a brief summary of PB and SG components in U-2 OS, relying for instance on a table (including references), with illustrating images in the supplemental data. (ii) I do not observe the shortening of this section announced by the authors (now 54 lines instead of 51). (iii) I am not convinced by the response relative to Figure S2A (absolute numbers of molecules would be more informative than relative abundance, since the point is not that DDX6 is more abundant than other PB proteins but that it can fulfill other functions besides making PBs). In fact, the text reads "DDX6 was shown to be much more abundant than other PB components in U-2 OS cells (Fig. S2 A)".

2/ Positioning with respect to the Majercki et al. preprint (BioRxiv 2021). Despite my mistake in the number of citations (1 instead of 2, both for elusive reasons), I find legitimate to wonder why, for the best information of the reader, there was no discussion relative to this very closely related study. The use of BioRxiv preprints is a delicate issue. It is claimed that the objective of these preprints is to give free and rapid access to the results of academic research before publication. If competing groups use it to run similar studies and publish first, it will kill the system. We have so many questions we can address in research! Anyway, the Majercki paper is now published and well discussed in the revised version.

4/(ii) The consequence of silencing DCP1A is still unclear to me. In Figure S4B, unfortunately, the pdf conversion prevents understanding the statistical analysis. Moreover I could not find a description of the first sample (--) of the graph (same for Figures 5B-D, S4, B-C, E-F). If it is a second type of negative control, then DCP1A KD does not "show a minor increase in PB numbers" (in line 326). Indeed, the average is 14 for siDCP1A vs 15 for --.

4/(iii) I regret the authors do not want to optimize the scale of the y-axis, as many differences seem of minor extent (even though statistically significant). I am also not aware that mean are more commonly used than median. Rather, their respective use should depend on the distribution of the data (symetric or not) and on the question.

4/(v) My point was about translation in the absence of stress (not after arsenite), since it has been shown that compounds that inhibit translation initiation (e.g. pateamine and hippuristanol) promote the formation of SGs (e.g. PMID: 16870703). As DDX6 KD diminishes translation before stress (new figure S2M, left panel), does the SG increase after stress results from a specific function of DDX6 with respect to SGs or from the decreased translation. In the latter case, any KD leading to decreased translation would similarly promote SGs, independently of the localization of the KD protein with respect to SGs.

4/(vi) I do not understand the answer of the authors, since I thought the graph shows individual cells. I do not see any evidence of two cell population (one silenced one not) that could temper the quantitatively minor effect.

5/(ii) I apologize for not being clear. I have no doubt that EDC4 is making punctate signal. However, when the number of foci reaches a very high density (say, over 300 per cell), a number of spots located below or above the granules should overlap with the granules and their periphery due to the maximum intensity projection used for quantification. In these conditions, how is it possible to accurately measure docking EDC4 spots, as shown in Figure S4E (left panel)?

5/(iii) I believe mentioning the debate about correlation between SG size and PB-SG docking (Aulas et al 2015) in the discussion would be of interest to the reader.

Reviewer #3 (Comments to the Authors (Required)):

With the substantial rewriting and improvement / addition of experimental data, this is now a very nice manuscript with a clear story that systematically analyses the effect of DDX6 KO or mutation on PB and SG. We have no major comments for revision, only two minor typos:

- line 73 misses a closing bracket after LSM14B
- line 257 double " in in"

Dear Dr. Parker,

Thank you for submitting your revised manuscript entitled "DDX6 modulates P-body and stress granule assembly, composition and docking". The manuscript has been seen by the original reviewers whose full comments are appended below. While the reviewers continue to be overall positive about the work in terms of its suitability for JCB, some important issues remain.

We thank the editor for the opportunity to address these final minor comments.

As you will see, while the reviewers appreciate that your study has been much improved a few important points remain. Regarding the level of advance, as your study was submitted to JCB before the Majerciak paper was published in a peer reviewed journal, our policy is that it does not detract from the novelty. However, while we appreciate that you have added in additional discussion of the Majerciak paper we agree that you should take the opportunity to discuss and synthesize prior work on this topic to present the reader with an overall current view. This seems especially important, since reviewer 1 mentions confusion about earlier literature regarding the idea that DDX6 is either more concentrated or exclusively localized to one or the other condensate (in particular, with reference to Kedersha and Anderson 2007). Perhaps you could discuss here that the 3D visualization in Fig 4 does not rule out presence of components on the inside of SGs.

As requested, we have added discussion about how the 3D imaging relates to possible inclusion of protein components within stress granules, and we have clarified references to earlier work.

In Figure S2J you must include untreated cells as requested.

We have done so.

Adding double staining with PABPC1 in S5E is not essential. Given your previous response to beginning your study with a recapitulation of what is mainly published data, we respect your choice of presentation. Finally, in your final revision please carefully respond to and address all other remaining reviewer concerns.

As detailed below, we have addressed the other reviewer comments.

Our general policy is that papers are considered through only one revision cycle; however, given that the suggested changes are relatively minor we are open to one additional short round of revision.

Please submit the final revision within one month, along with a cover letter that includes a

point by point response to the remaining reviewer comments.

Thank you for this interesting contribution to Journal of Cell Biology. You can contact me or the scientific editor listed below at the journal office with any questions at cellbio@rockefeller.edu.

Sincerely,

Karla Neugebauer, PhD
Monitoring Editor

Andrea L. Marat, PhD
Senior Scientific Editor

Journal of Cell Biology

Reviewer #1 (Comments to the Authors (Required)):

Overall, the manuscript has been much improved by the revisions. The manuscript is much clearer now in terms of defining what constitutes a SG or PB, and the rationale for the markers they use. The authors also included several experiments proposed by the reviewers which strengthen their claims. The manuscript will be strengthened by a little more discussion of how this work complements and/or contradicts the Majerciak paper, as well as addressing a few other minor comments.

- While the authors have added more discussion on Majerciak, one of the key points is not mentioned, which is that they propose that DDX6 is essential for the separation (creating the normal docking) of SGs and PBs. It would be appropriate to mention this in the discussion paragraph on docking.

We have noted that our conclusion about DDX6 affecting PB-SG docking is similar to that of Majerciak and cited the manuscript appropriately.

- Fig S2J is a good addition, however it is so small it is hard to see what they authors are hoping to show with it. It would also help to include the same call line but untreated (without arsenite). I think this is important here because the assemblies are not as distinct as in WT cells, so comparing the images to untreated cells may help make the point stronger.

We have added the requested images in untreated conditions and expanded the size of the image as requested.

- Fig S5 E: including PABPC1 here would make it easier for the reader to compare to the corresponding figure in the main text (Fig 5). Please also increase the brightness of the image so EDC4 is clearer.

We appreciate the editorial decision to allow this figure to remain without additional PABPC1 immunofluorescence.

However, we have not increased the brightness of EDC4 any further because doing so leads to an overexposed-looking image and larger overlapping EDC4 spots. The EDC4 spots start to appear as big, merged blobs instead of spots, which does not reflect the actual protein localization.

- Fig 4 - 3D visualization clearly shows that PBs are on the outside of SGs, but does it rule out their presence inside?

We agree with the reviewer that the presence of PBs on the outside of SG does not rule out internal localization as well, which we had already described in the text "the majority of EDC4 and DCP1A puncta are indeed docking to the SG surface, with only a few puncta that can be seen inside" and "4E-T protein are more heterogeneous and located at the SG center and inside".

- Confusion over whether DDX6 is in PBs or not: Line 141-142: this is incorrect. Kedersha and Anderson 2007 show greater DDX6 (P54/RCK) recruitment to PBs than SGs, but not that it is only in PBs. Line 161-162 supports DDX6 in SGs.

We clarified this point in the text. We agree with the reviewer that there is no dispute that earlier manuscripts have shown DDX6 can be in both PB and SG. The issue is that there are differences between the text of the Kedersha and Anderson 2007 manuscript, which describes DDX6 as being in both PB and SG, and the figure legends of the same manuscript that state "Hedls and p54-RCK (which is DDX6) stain PBs exclusively, whereas eIF3 is specific for SGs." In addition, the images (Figure 5.1+5.2) in the manuscript show no SG localization of DDX6 in PBs. We now believe this issue is clarified in the text.

- Line 262 - 263: should this be "enrichment in SGs and PBs in WT cells"?

We have clarified this sentence.

- Line 275: components, not comments

Corrected.

- Should be Brigham and Women's Hospital

Corrected.

Reviewer #2 (Comments to the Authors (Required)):

The authors have now appropriately cited the previous literature, including the Majerciak et al. recent publication. Compared to some of this work, I agree with the authors that the quality of imaging is increased by the use of newer microscopes. Yet, I believe the novelty of the conclusions is too incremental for the Journal of Cell Biology. I also maintain it is difficult to build a strong model based only on variations of SG size and PB-SG docking, particularly when these variations are quantitatively minor.

Some of my previous concerns were not fully addressed:

1/ Beginning of the results that essentially replicate published data. (i) I agree with the authors that they need to verify the presence of the various proteins they will study in SG and in PBs in U-2 OS cells, since differences between cell lines were reported in the past. However, for the reader who is eager to see new data, it would be more efficient to give a brief summary of PB and SG components in U-2 OS, relying for instance on a table (including references), with illustrating images in the supplemental data.

While we appreciate the reviewer's comment, we believe showing all these IFs in one place for all the markers at the same time will be a valuable contribution and would be less so as a Table, which will still take up room. Given this, we appreciate the editorial decision to allow this Figure to remain in the manuscript.

(ii) I do not observe the shortening of this section announced by the authors (now 54 lines instead of 51).

We apologize for this error in our previous letter.

(iii) I am not convinced by the response relative to Figure S2A (absolute numbers of molecules would be more informative than relative abundance, since the point is not that DDX6 is more abundant than other PB proteins but that it can fulfill other functions besides making PBs). In fact, the text reads "DDX6 was shown to be much more abundant than other PB components in U-2 OS cells (Fig. S2 A)".

We agree with the reviewer the key point is indeed that DDX6 is much more abundant than other PB components in U-2 OS cells, and thereby might have alternative functions. We have now clarified that point in the text.

2/ Positioning with respect to the Majerciak et al. preprint (BioRxiv 2021). Despite my mistake in the number of citations (1 instead of 2, both for elusive reasons), I find legitimate to wonder why, for the best information of the reader, there was no discussion relative to this very closely related study. The use of BioRxiv preprints is a delicate issue. It is claimed that the objective of these preprints is to give free and rapid access to the results of academic research before publication. If competing groups use it to run similar studies and publish first, it will kill the system. We have so many questions we can address in research! Anyway, the Majerciak paper is now published and well discussed in the revised version.

We understand the reviewer's concern. However, we do point out we have been studying this area since 2019 and did not begin our work because of the Bioarchive manuscript from Majerciak. Indeed, we reached out to them and shared all of our data with them at the RNA 2022 Society conference, which we believe heavily influenced their final manuscript. This is how science should work where groups synergize as much as possible.

4/(ii) The consequence of silencing DCP1A is still unclear to me. In Figure S4B, unfortunately, the pdf conversion prevents understanding the statistical analysis. Moreover I could not find a description of the first sample (--) of the graph (same for Figures 5B-D, S4, B-C, E-F). If it is a second type of negative control, then DCP1A KD does not "show a minor increase in PB numbers" (in line 326). Indeed, the average is 14 for siDCP1A vs 15 for --.

We apologize for not explicitly stating that the "-" samples are stressed cells which did not receive any transfection, which we now clarify in the figure legends. The statistics here are done between non-targeting siRNA control and a targeting siRNA.

4/(iii) I regret the authors do not want to optimize the scale of the y-axis, as many differences seem of minor extent (even though statistically significant). I am also not aware that mean are more commonly used than median. Rather, their respective use should depend on the distribution of the data (symetric or not) and on the question.

We agree that different methods of presentation can be more effective, but the critical issue is whether a change is significant in an unbiased analysis, which we have used in all our comparisons.

4/(v) My point was about translation in the absence of stress (not after arsenite), since it has been shown that compounds that inhibit translation initiation (e.g. pateamine and hippuristanol) promote the formation of SGs (e.g. PMID: 16870703). As DDX6 KD diminishes translation before stress (new figure S2M, left panel), does the SG increase after stress results from a specific function of DDX6 with respect to SGs or from the decreased translation. In the latter case, any KD leading to decreased translation would similarly promote SGs, independently of the localization of the KD protein with respect to SGs.

In this comment, the reviewer is raising the possibility that differences in translation in the DDX6Δ cells might alter the number of stress granules formed. We agree this is a formal possibility, and this is why we examined translation rates in both WT and DDX6Δ cells during the stress response (when stress granules are forming) by puromycin labeling (Figure S2L/M). Since we observe that all the cell line repress translation similarly, we interpret this to suggest that differences in SG formation between WT and DDX6Δ cells are not due to differences in translation in stressed conditions.

4/(vi) I do not understand the answer of the authors, since I thought the graph shows individual cells. I do not see any evidence of two cell population (one silenced one not) that could temper the quantitatively minor effect.

It is clear from our western blots (Figure S4A) that some of our siRNAs are not 100%, given this we are almost certainly looking at a mixed population of cells with phenotypes that overlap in their distribution. This provides a plausible explanation for the smaller phenotypic difference, which is reflected in our interpretations.

5/(ii) I apologize for not being clear. I have no doubt that EDC4 is making punctate signal. However, when the number of foci reaches a very high density (say, over 300 per cell), a number of spots located below or above the granules should overlap with the granules and their periphery due to the maximum intensity projection used for quantification. In these conditions, how is it possible to accurately measure docking EDC4 spots, as shown in Figure S4E (left panel)?

We agree with the reviewer that there could be a potential complication with quantifying images using maximum intensity projections. However, all of our analyses here are done using the Image Analysis Software (Bitplane) either on the 3D image (where spots above or below the granule are detected being outside) or on a single plane (both of which give similar results). To clarify this method, we have added a reference for this approach in the methods (Khong et al., 2018, Methods).

5/(iii) I believe mentioning the debate about correlation between SG size and PB-SG docking (Aulas et al 2015) in the discussion would be of interest to the reader.

We would prefer not to add a discussion of the data in Aulas et al., 2015 for the following reasons. First, the main observation in Aulas et al 2015 is that knockdowns of specific proteins show changes in both SG size and % of PB docking. However, the authors did not examine whether the change in PB docking is due to the size of the SG or the reduction in the specific proteins. In our work, when we examine SGs that are docked to PBs, they do not show an average increase in size, which argues that the difference in docking seen with these perturbations is due to the specific proteins per se and not the size of the PB or SG.

We have not included this analysis in our manuscript since it is complicated to explain and really is addressing a minor possibility, and we provide significant other data arguing for protein-protein interactions affecting docking. If you as editor felt it was important, we could add this data to the manuscript, and an additional discussion about the relationship of granule size to docking, but we believe this would complicate rather than improve the manuscript.

Reviewer #3 (Comments to the Authors (Required)):

With the substantial rewriting and improvement / addition of experimental data, this is now a very nice manuscript with a clear story that systematically analyses the effect of DDX6 KO or mutation on PB and SG. We have no major comments for revision, only two minor typos:

- line 73 misses a closing bracket after LSM14B
- line 257 double " in in"

We have corrected both typos as requested.

February 19, 2024

RE: JCB Manuscript #202306022RR

Dr. Roy Parker
University of Colorado Boulder
Dept of Biochemistry
JSCBB
Boulder, CO 80309

Dear Dr. Parker:

Thank you for submitting your revised manuscript entitled "DDX6 modulates P-body and stress granule assembly, composition and docking". We appreciate your final revisions and agree that your study provides an interesting contribution to our understanding of the relationship between stress granules and P-bodies as well as the composition and function of the two. Therefore, we would be happy to publish your paper in JCB pending final revisions necessary to meet our formatting guidelines (see details below).

A. MANUSCRIPT ORGANIZATION AND FORMATTING:

- 1) Text limits: Character count for Articles is < 40,000, not including spaces. Count includes abstract, introduction, results, discussion, and acknowledgments. Count does not include title page, figure legends, materials and methods, references, tables, or supplemental legends.
- 2) Figures limits: Articles may have up to 10 main text figures.
- 3) Figure formatting: Scale bars must be present on all microscopy images, including all inset magnifications. Molecular weight or nucleic acid size markers must be included on all gel electrophoresis.
- 4) Statistical analysis: Error bars on graphic representations of numerical data must be clearly described in the figure legend. The number of independent data points (n) represented in a graph must be indicated in the legend. Statistical methods should be explained in full in the materials and methods. For figures presenting pooled data the statistical measure should be defined in the figure legends. Please also be sure to indicate the statistical tests used in each of your experiments (either in the figure legend itself or in a separate methods section) as well as the parameters of the test (for example, if you ran a t-test, please indicate if it was one- or two-sided, etc.). Also, if you used parametric tests, please indicate if the data distribution was tested for normality (and if so, how). If not, you must state something to the effect that "Data distribution was assumed to be normal but this was not formally tested."
- 5) Abstract and title: The abstract should be no longer than 160 words and should communicate the significance of the paper for a general audience. The title should be less than 100 characters including spaces. Make the title concise but accessible to a general readership.
- 6) Materials and methods: Should be comprehensive and not simply reference a previous publication for details on how an experiment was performed. Please provide full descriptions in the text for readers who may not have access to referenced manuscripts.
- 7) All antibodies, cell lines, animals, and tools used in the manuscript should be described in full, including accession numbers for materials available in a public repository such as the Resource Identification Portal. Please be sure to provide the sequences for all of your primers/oligos and RNAi constructs in the materials and methods. You must also indicate in the methods the source, species, and catalog numbers (where appropriate) for all of your antibodies. Please also indicate the acquisition and quantification methods for immunoblotting/western blots.
- 8) Microscope image acquisition: The following information must be provided about the acquisition and processing of images:
 - a. Make and model of microscope
 - b. Type, magnification, and numerical aperture of the objective lenses
 - c. Temperature

- d. Imaging medium
- e. Fluorochromes
- f. Camera make and model
- g. Acquisition software
- h. Any software used for image processing subsequent to data acquisition. Please include details and types of operations involved (e.g., type of deconvolution, 3D reconstitutions, surface or volume rendering, gamma adjustments, etc.).

10) Supplemental materials: There are strict limits on the allowable amount of supplemental data. Articles may have up to 5 supplemental figures. Please also note that tables, like figures, should be provided as individual, editable files. A summary of all supplemental material should appear at the end of the Materials and methods section.

13) ORCID IDs: ORCID IDs are unique identifiers allowing researchers to create a record of their various scholarly contributions in a single place. Please note that ORCID IDs are now *required* for all authors. At resubmission of your final files, please be sure to provide your ORCID ID and those of all co-authors.

Please note that JCB now requires authors to submit Source Data used to generate figures containing gels and Western blots with all revised manuscripts. This Source Data consists of fully uncropped and unprocessed images for each gel/blot displayed in the main and supplemental figures. Since your paper includes cropped gel and/or blot images, please be sure to provide one Source Data file for each figure that contains gels and/or blots along with your revised manuscript files. File names for Source Data figures should be alphanumeric without any spaces or special characters (i.e., SourceDataF#, where F# refers to the associated main figure number or SourceDataFS# for those associated with Supplementary figures). The lanes of the gels/blots should be labeled as they are in the associated figure, the place where cropping was applied should be marked (with a box), and molecular weight/size standards should be labeled wherever possible.

Journal of Cell Biology now requires a data availability statement for all research article submissions. These statements will be published in the article directly above the Acknowledgments. The statement should address all data underlying the research presented in the manuscript. Please visit the JCB instructions for authors for guidelines and examples of statements at (<https://rupress.org/jcb/pages/editorial-policies#data-availability-statement>).

B. FINAL FILES:

****It is JCB policy that if requested, original data images must be made available to the editors. Failure to provide original images upon request will result in unavoidable delays in publication. Please ensure that you have access to all original data images prior to final submission.****

****The license to publish form must be signed before your manuscript can be sent to production. A link to the electronic license to publish form will be sent to the corresponding author only. Please take a moment to check your funder requirements before choosing the appropriate license.****

Thank you for this interesting contribution, we look forward to publishing your paper in Journal of Cell Biology.

Sincerely,

Karla Neugebauer, PhD
Monitoring Editor

Andrea L. Marat, PhD
Senior Scientific Editor

Journal of Cell Biology